# AAA+ protease-adaptor structures reveal altered conformations and ring specialization

Sora Kim [1,2,4], Xue Fei[1,3,4], Robert T. Sauer [1] and Tania A. Baker [1] ✉

ClpAP, a two-ring AAA+ protease, degrades N-end-rule proteins bound by the ClpS adaptor. Here we present high-resolution cryo-EM structures of *Escherichia coli* ClpAPS complexes, showing how ClpA pore loops interact with the ClpS N-terminal extension (NTE), which is normally intrinsically disordered. In two classes, the NTE is bound by a spiral of pore-1 and pore-2 loops in a manner similar to substrate-polypeptide binding by many AAA+ unfoldases. Kinetic studies reveal that pore-2 loops of the ClpA D1 ring catalyze the protein remodeling required for substrate delivery by ClpS. In a third class, D2 pore-1 loops are rotated, tucked away from the channel and do not bind the NTE, demonstrating asymmetry in engagement by the D1 and D2 rings. These studies show additional structures and functions for key AAA+ elements. Pore-loop tucking may be used broadly by AAA+ unfoldases, for example, during enzyme pausing/unloading.

Regulated proteolysis by energy-dependent AAA+ (ATPases associated with diverse cellular activities) proteases maintains protein homeostasis in all organisms. AAA+ proteases degrade regulatory proteins and proteins that are damaged, misfolded or no longer needed[1]. AAA+ proteases, such as ClpAP, consist of a hexameric AAA+ unfoldase (for example, ClpA$_6$) and a self-compartmentalized peptidase (for example, ClpP$_{14}$). In the recognition step, a peptide sequence (degron) in a protein substrate is engaged by pore loops lining the unfoldase channel. Through conformational changes powered by ATP-hydrolysis cycles, native structure in the bound substrate is unfolded and processively translocated through the channel and into the peptidase chamber, where the polypeptide is cleaved. In addition to binding and engaging degrons directly, AAA+ proteases interact with adaptor proteins that modify their substrate specificity[1,2].

Prokaryotes and eukaryotes use the N-end-rule pathway to target proteins bearing specific N-terminal residues (called N-degrons) for rapid degradation[3]. In *Escherichia coli*, the ClpS adaptor promotes ClpAP degradation of proteins containing Leu, Phe, Tyr or Trp residues at the N-terminus[3-8]. ClpS (~10 kDa) docks with the N-terminal domain of ClpA and contains a hydrophobic pocket that binds the N-end-rule residue[6-10]. ClpS functions as a specificity switch for ClpAP, promoting degradation of N-degron substrates while inhibiting degradation of ssrA-tagged and related substrates[5,8,9,11]. Interestingly, ClpS is proposed to interact with ClpA as a 'pseudo-substrate'[5,12-15]. Specifically, the N-terminal extension (NTE) of free ClpS is exposed as an unstructured peptide, mimicking a degron. The NTE is poorly conserved among orthologs, with the exception of a short junction sequence adjacent to the ClpS core that typically contains a few tandem prolines[13,16]. During ClpS-assisted degradation, a ClpS•N-degron substrate complex initially binds to ClpA. Subsequently, the N-degron substrate is transferred to ClpA for degradation, and ClpS escapes destruction by mechanisms that are poorly understood.

Each ClpA subunit has two AAA+ modules, called D1 and D2, that associate in the hexamer to form two stacked rings[17]. The D1 and D2 modules belong to different AAA+ subfamilies and have distinct biochemical functions[18]. The D2 ring, a member of the HCLR AAA+ clade, is the principal ATPase motor responsible for unfolding and translocating substrates, including proteins with high thermodynamic stabilities[19-21]. In contrast, the D1 ring, a classic AAA+ clade member, assists the D2 ring as an auxiliary motor, improves enzyme processivity

[1]Department of Biology, Massachusetts Institute of Technology, Cambridge, MA, USA. [2]Present address: Department of Microbiology, University of Washington, Seattle, WA, USA. [3]Present address: Kymera Therapeutics Inc., Watertown, MA, USA. [4]These authors contributed equally: Sora Kim, Xue Fei. ✉e-mail: tabaker@mit.edu

and plays a major role in substrate recognition[19–22]. ClpS differentially regulates the activities of the D1 and D2 rings[19,22] via interactions of its NTE, which we characterize here. Previous cryo-EM structures of ClpAP elucidate how the axial channel of the D1 and D2 rings engages the polypeptide of a directly recognized substrate[23]. Pore-1 and pore-2 loops in both rings form spiral-staircase-like arrangements that bind the substrate polypeptide, in a similar manner to those in structures of other double-ring AAA+ enzymes, such as Hsp104, ClpB, Cdc48/p97 and N-ethylmaleimide-sensitive factor (NSF)[24–31]. However, these previous structures do not provide insight into the specialized functions of each ClpA AAA+ ring or the mechanism of ClpS-assisted degradation of N-degron substrates.

To characterize ring specialization and ClpS-ClpA collaboration, we solved cryo-EM structures of ClpAPS complexes that show how the normally disordered ClpS NTE assumes an extended conformation when bound in the ClpA channel. These structures reveal marked conformational differences from prior structures[23]. We identify multiple conformations of ClpS-bound ClpA, including an arrangement in which the D2 ring pore-1 loops are tucked in and face away from the channel, allowing only the D1 ring to interact strongly with the ClpS NTE. Mutagenesis and biochemical experiments establish that the ClpA D1 ring pore-2 loops are essential for ClpS delivery of an N-degron substrate, but contribute little to docking of ClpS with ClpA. Our results demonstrate structural and functional plasticity among ClpA pore loops, provide a structural basis for the functions of ClpS during N-degron substrate degradation and contribute more broadly to understanding the operational modes available to AAA+ enzymes when performing diverse biological processes.

## Results

### Distinct conformations of ClpAPS delivery complexes

We used size-exclusion chromatography in the presence of ATPγS to purify a complex of ClpA, ClpP, ClpS and the N-end-rule substrate YLFVQELA-GFP (Fig. 1a and Supplementary Fig. 1). Based on sodium dodecyl sulfate polyacrylamide gel electrophoresis (SDS–PAGE), the YLFVQELA-GFP substrate appeared to be sub-stoichiometric compared to ClpS (Fig. 1b). Because ATPγS does not support degradation[23,32–36], these complexes should represent early stages in ClpS-mediated delivery of N-degron substrates.

Following single-particle cryo-EM analyses (Extended Data Fig. 1), three-dimensional (3D) classification and reconstruction using RELION-3 yielded six density maps (3.22–3.38 Å), representing three general structural classes (I, II and III), with the latter classes being subdivided into $II_a/II_b/II_c$ or $III_a/III_b$ subclasses (Table 1, Extended Data Fig. 2 and Supplementary Figs. 2–4). In low-pass filtered maps, the ClpS core domain (res. 27–106) could be docked into each map (Extended Data Fig. 3). In unfiltered maps, there was good density for all or part of the NTE of ClpS, for the D1 and D2 rings of ClpA and for ClpP (Fig. 1c,d). There was no substantial density for the ClpS core, ClpA N-domains, or YLFVQELA-GFP, suggesting that these domains/proteins are not present in fixed conformations relative to the remaining parts of the complex or are potentially absent (YLFVQELA-GFP). Given the lack of observed substrate density, it is possible that some of our structures represent ClpS-only complexes at an earlier assembly step, and differences between our classes could be due to the presence or absence of substrate. However, because ClpS binds ClpA -ninefold more weakly without substrate[10], under our purification conditions, the prepared sample most likely contains higher-affinity ClpS•substrate-bound complexes.

In our structures, the six subunits of the ClpA hexamer, which we label A through F (clockwise direction with subunit F at the bottom), formed a shallow spiral, as expected from earlier cryo-EM structures[23]. Six flexible Ile-Gly-Leu (IGL) loops (res. 610–628) in each ClpA hexamer docked into clefts in the heptameric ClpP ring, leaving one empty cleft between those occupied by subunits E and F (Fig. 1b). Differences

between classes I, II and III include the relative positions of subunits in the ClpA spiral, density for the ClpS NTE in the ClpA channel and changes within individual ClpA subunits (Figs. 1c,d and 2 and Supplementary Figs. 4 and 5). For example, density for the ClpS NTE was present in both the ClpA D1 and D2 rings in classes I and II, but only in the D1 ring of class III (Fig. 1d). In classes II and III, the relative height of ClpA subunits was A (highest) > B > C > D > E > F (lowest), whereas in class-I, subunit B was higher than subunit A, resulting in a shallower spiral when aligned to the bottom of the IGL loop and the two flanking ClpP subunits (Fig. 2a). Additionally, in the D2 ring of class-III structures, the small AAA+ domain of subunit E (res. 656–749) swings outward from the hinge-linker (res. 650–655), breaking the rigid-body interface with its large AAA+ domain neighbor (res. 442–649) from subunit F (Fig. 2b,c). The subclasses ($II_a/II_b/II_c$ or $III_a/III_b$) differed from each other largely in the detailed interactions between ClpA and the ClpS NTE, the visibility of individual NTE residues and the nucleotide occupancy of each ATPase site (ATPγS, ADP or empty; Extended Data Figs. 4–7 and Supplementary Fig. 5).

### Binding of the ClpS NTE within the axial channel of ClpA

Each of our structures contained clear main-chain and side-chain density corresponding to all or part of the ClpS NTE (res. 2–26) in the ClpA channel (Fig. 3a and Supplementary Fig. 5). The register of this ClpS peptide is very similar in each structure, with the C-terminal portion of the NTE ($Pro^{24}–Pro^{25}–Ser^{26}$) near the top of the ClpA channel, and the N-terminal portion near the bottom of the channel in classes I and II. The ClpA channel is lined by the D1 KYR pore-1 loops (res. 258–260) and pore-2 loops (res. 292–302) and by the D2 GYVG pore-1 loops (res. 539–542) and pore-2 loops (res. 526–531). Pore-1 loops of AAA+ unfoldases and protein-remodeling machines contain a key, conserved aromatic side chain (usually tyrosine; underlined in K$\underline{Y}$R and G$\underline{Y}$VG) that contacts the substrate polypeptide in the channel and functions in the binding, unfolding and translocation of target proteins[22,23,37–42]. The ClpS NTE was bound by many K$\underline{Y}$R and G$\underline{Y}$VG pore-1 loops and also by the D1 pore-2 loops. Neighboring pore-1 loops interacted with two-residue segments of the NTE, as observed for substrate polypeptides bound to multiple AAA+ unfoldases and protein-remodeling machines[43].

Despite this overall resemblance to substrate engagement, there were deviations in individual pore-1 loop interactions from those in previous structures of ClpAP and some Hsp100 family members. For example, the D2 G$\underline{Y}$VG pore-1 loops of all six ClpA subunits contacted the ClpS NTE in classes I and $II_c$, albeit the D2 pore-1 loop of subunit A had a lower buried surface area (BSA) in class I than in $II_c$ (Extended Data Figs. 6 and 8). Previous ClpAP structures and subclasses $II_a$ and $II_b$ show four or five engaged G$\underline{Y}$VG loops (Extended Data Fig. 8b)[23]. The configuration of pore-1 loops in classes I and $II_c$ was also different from an extended Hsp104•casein structure in which loops from both the top and bottom AAA+ rings of all six protomers contact the substrate in a split 'lock-washer' conformation[25]. In classes I and $II_c$, we observed five bound pore-1 loops and one unbound pore-1 loop in D1 and six bound pore-1 loops in D2, an arrangement found in the high-affinity state of *Mycobacterium tuberculosis* ClpB[26]. In many AAA+ structures, only the pore loops of ATP-bound subunits contact the substrate[43]. By contrast, and as reported for ClpAP•substrate complexes[23], the pattern of engaged versus disengaged pore loops in our structures did not strictly correlate with the nucleotide present in the corresponding ATPase active site (Extended Data Figs. 4 and 5). For instance, ADP is bound to the class-$II_c$ D2 nucleotide sites in subunits E and F, but the G$\underline{Y}$VG loops from these domains contact the NTE. The presence of 11 engaged pore-1 loops (five D1 and six D2) probably contributes to the high affinity of ClpAPS•N-degron complexes assembled in ATPγS[13].

### D2 pore-1 loops rotate outward to alter polypeptide contacts

In classes I and II, residues 2–15 of the ClpS NTE were built into density in the D2 portion of the channel, but this NTE region was not visible in

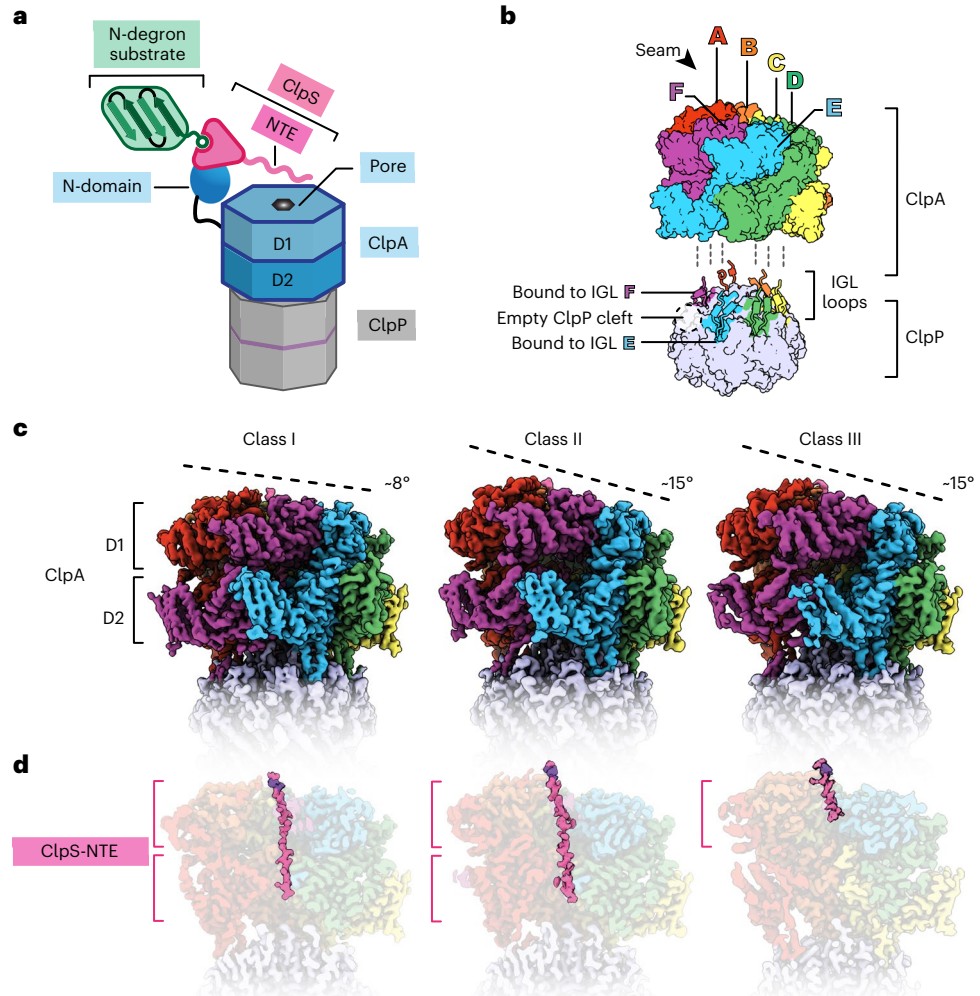

**Fig. 1 | Architectures of ClpS-bound ClpAP. a**, Cartoon of ClpS delivery of an N-degron substrate for ClpAP degradation. Native ClpS (pink wedge) binds to the N-degron substrate (green) and also binds to an N-terminal domain of ClpA (blue oval). **b**, ClpA subunit nomenclature in the right-hand spiral hexamer, where the seam interface is between the lowest (F) subunit and the highest (A) subunit. The ClpA hexamer docks into clefts in the ClpP$_7$ ring via IGL loops. The empty ClpP cleft is located between the clefts occupied by subunits E and F. **c**, Side views of the cryo-EM maps of classes I, II$_c$ and III$_b$. The dashed line indicates the relative height of ClpA subunits within the spiral, aligned to a common view using ClpP$_7$. **d**, Cutaway views of **c** showing density for the ClpS NTE colored pink.

class III, presumably as a consequence of its conformational heterogeneity. We infer that these NTE residues are within D2, as the more C-terminal NTE segment (res. 16–26) is bound by the D1 ring of class III in the same manner as in classes I and II. Thus, the two AAA+ rings of ClpA can differ in their engagement with the NTE, a feature not observed in substrate-bound ClpAP structures[23].

This absence of density for the N-terminal portion of the NTE in class III correlated with distinct structural features within the axial channel. Most surprisingly, the D2 pore-1 loops in class III were rotated ~90° compared to their orientation in classes I and II, and the key Tyr[540] side chains were tucked in and turned away from the channel (Fig. 3b, Extended Data Fig. 7 and Supplementary Video 1). In both class-III subclasses, at least four of the six pore-1 loops were convincingly in the tucked conformation. In many AAA+ unfoldases and protein-remodeling machines, one or two pore-1 loops, usually at the top and bottom of the spiral, are disengaged from the substrate polypeptide as a result of translational displacement of the corresponding subunit(s)[23–31,43–52]. This 'canonical' disengaged state of pore-1 loops in one or two subunits differs very much from the tucked and rotated orientations of the class-III D2 pore-1 loops, in which no interactions with the polypeptide in the channel were present in the entire D2 ring.

Pore-1 tyrosine contacts with the polypeptide within the AAA+ channel are considered essential for substrate binding and translocation. Thus, rotation of most (or all) Tyr[540] side chains in the class-III D2 ring is sufficient to explain the lack of initial engagement of the N-terminal segment of the NTE and/or loss of binding that may occur during ClpS-assisted degradation of N-degron substrates (see Discussion).

Three additional features of the class-III D2 ring are noteworthy. Coincident with the pore-1-loop rotation, the ClpA channel in the D2 ring of class III was ~2 Å wider than in classes I and II (Fig. 2c and Supplementary Fig. 6). Second, as noted above, the D2 rigid-body interface between the small AAA+ domain of subunit E and its neighboring large AAA+ domain in subunit F was broken in class III. This rearrangement may facilitate the accompanying conformational changes that result in loss of NTE contacts by the D2 pore-1 loops. Finally, the D2 ring contained ADP in three adjacent subunits in class III, whereas classes I and II contained no more than two ADPs in the D2 ring (Extended Data Fig. 5). Thus, ClpA has the ability to tightly bind all of the NTE polypeptide in the axial channel using pore loops in both rings or to bind only the C-terminal portion of this sequence within the D1 ring, suggesting that the polypeptide-binding activities of the D1 and D2 rings can be either coordinated or independent.

**Table 1 | Cryo-EM data collection, processing, model building and validation statistics**

| | I (EMDB-26556, PDB 7UIX) | II$_a$ (EMDB-26554, PDB 7UIV) | II$_b$ (EMDB-26555, PDB 7UIW) | II$_c$ (EMDB-26558, PDB 7UIZ) | III$_a$ (EMDB-26557, PDB 7UIY) | III$_b$ (EMDB-26559, PDB 7UJO) |
|---|---|---|---|---|---|---|
| **Data collection and processing** | | | | | | |
| Microscope | Talos Arctica | | | | | |
| Camera | K3 | | | | | |
| Nominal magnification | ×45,000 | | | | | |
| Voltage (kV) | 200 | | | | | |
| Data acquisition software | SerialEM | | | | | |
| Exposure navigation | Image shift to four holes | | | | | |
| Total electron dose (e⁻/Å$^2$) | 34.71 | | | | | |
| Exposure rate (e⁻/pixel/s) | 5.1 | | | | | |
| Number of frames per micrograph | 26 | | | | | |
| Defocus range (μm) | −0.5 to −2.5 | | | | | |
| Pixel size (Å) | 0.87 | | | | | |
| Micrographs collected | 9,169 | | | | | |
| Total extracted particles | 1,043,033 | | | | | |
| Refined particles | 717,833 | | | | | |
| Final particles | 51,750 | 156,677 | 43,431 | 37,530 | 37,885 | 31,453 |
| Symmetry | $C1$ | $C1$ | $C1$ | $C1$ | $C1$ | $C1$ |
| Map resolution (Å) at 0.143 FSC | 3.24 | 3.38 | 3.33 | 3.24 | 3.22 | 3.26 |
| Local resolution range (Å) | 3–9 | 3–9 | 3–8 | 3–10 | 3–8 | 3–10 |
| Accuracy of translations | 0.40 | 0.60 | 0.42 | 0.40 | 0.40 | 0.40 |
| Accuracy of rotations | 0.53 | 0.78 | 0.58 | 0.48 | 0.49 | 0.49 |
| Map sharpening $B$ factor (Å$^2$) | −91.3 | −107.8 | −88.5 | −76.6 | −80.5 | −86.7 |
| **Model composition** | | | | | | |
| Nonhydrogen atoms | 38,021 | 37,900 | 37,776 | 38,131 | 37,880 | 38,006 |
| Protein residues | 4,820 | 4,805 | 4,792 | 4,837 | 4,819 | 4,823 |
| Nucleotides | 12 | 11 | 10 | 12 | 12 | 12 |
| **Refinement** | | | | | | |
| Refinement package | Phenix | Phenix | Phenix | Phenix | Phenix | Phenix |
| Model resolution (Å) at 0.5 FSC unmasked | 3.4 | 3.5 | 3.4 | 3.4 | 3.3 | 3.4 |
| Model resolution (Å) at 0.5 FSC masked | 3.4 | 3.4 | 3.4 | 3.4 | 3.3 | 3.4 |
| Model resolution (Å) at 0.143 FSC unmasked | 2.8 | 2.9 | 2.9 | 2.9 | 2.8 | 2.9 |
| Model resolution (Å) at 0.143 FSC masked | 2.8 | 2.9 | 2.9 | 2.9 | 2.8 | 2.9 |
| Map–model CC (mask) | 0.80 | 0.77 | 0.75 | 0.81 | 0.81 | 0.82 |
| Map–model CC (volume) | 0.77 | 0.74 | 0.72 | 0.79 | 0.78 | 0.80 |
| R.m.s.d. bond lengths (Å) | 0.005 | 0.005 | 0.005 | 0.005 | 0.006 | 0.006 |
| RMSD bond angles (°) | 1.115 | 1.117 | 1.100 | 1.095 | 1.156 | 1.142 |
| $B$-factor protein (Å$^2$) | 61.3 | 67.9 | 63.9 | 74.6 | 72.1 | 88.0 |
| $B$-factor ligand (Å$^2$) | 55.5 | 61.9 | 42.4 | 65.0 | 66.3 | 86.1 |
| **Validation** | | | | | | |
| MolProbity score | 1.09 | 1.06 | 1.06 | 1.07 | 1.14 | 1.10 |
| Clash score | 2.98 | 2.74 | 2.74 | 2.78 | 3.48 | 3.05 |
| Cβ deviation (%) | 0.00 | 0.00 | 0.00 | 0.00 | 0.00 | 0.00 |
| Rotamer outliers (%) | 0.00 | 0.00 | 0.02 | 0.00 | 0.00 | 0.00 |
| CaBLAM outliers (%) | 1.35 | 1.48 | 1.68 | 1.53 | 1.66 | 1.93 |
| Ramachandran favored (%) | 99.94 | 99.73 | 99.75 | 99.88 | 99.77 | 99.50 |
| Ramachandran disallowed (%) | 0.00 | 0.00 | 0.00 | 0.00 | 0.00 | 0.00 |
| EMRinger score | 2.80 | 2.52 | 2.41 | 2.98 | 3.14 | 3.29 |

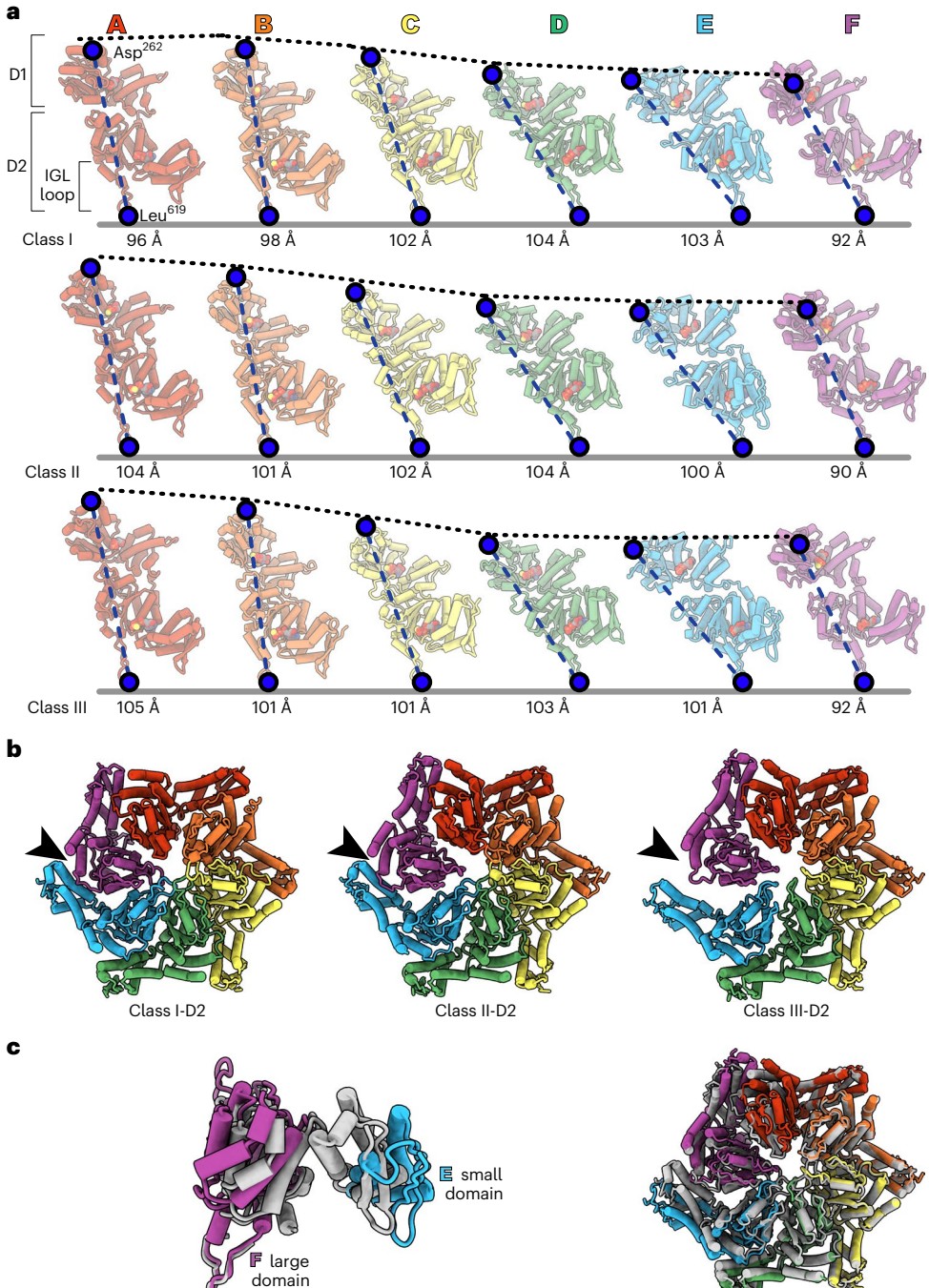

**Fig. 2 | Conformational differences in ClpA subunits and hexamers. a**, Individual subunits of class I, II$_c$ and III$_a$ atomic models. The dashed black lines indicate the relative height of each subunit, following alignment to the bottom of the IGL loop and the two flanking ClpP subunits (containing the ClpP cleft). The distance between Asp$^{262}$ and Leu$^{619}$ (indicated by blue circles) is shown below. **b**, D2-ring rigid-body interface between subunits E and F of classes I, II$_c$ and III$_a$.

In class III$_a$, the small AAA+ domain of subunit E in the D2 ring swings out and loses contact (arrow) with the neighboring large D2 AAA+ domain of subunit F. **c**, Overlay of the large (res. 444–649) and small (res. 655–749) AAA+ domains in class I (colored gray) and class III (colored by subunit). The pairwise Cα r.m.s.d. of the subunit E/F rigid-body interface (left panel) between classes I and III$_a$ is ~6.5 Å and between classes I and II$_c$ is ~2.9 Å.

## D1 pore-2 loops form a second network of NTE contacts

In addition to the pore-1 loop interactions described above, our structures show that at least four pore-2 loops (res. 292–302) in the D1 ring of ClpA contacted the ClpS NTE (Fig. 4). In each subunit, these pore-2 contacts were positioned below the corresponding D1 KYR contacts and were offset by ~60°. The Ala$^{295}$–Ala$^{296}$–Ser$^{297}$ tripeptide (AAS) at the tip of the D1 pore-2 loops contacted the opposing face of the ClpS NTE compared to the contacts made by the D1 pore-1 loops (compare the

orientation of the D1 pore-2 loops on the left versus the D1 pore-1 loops on the right side of the channel in Fig. 4a). In contrast to the well-defined KYR motif in the D1 pore-1 loop, which is conserved among Hsp104/ClpABC protein-remodeling enzymes and contains the invariant aromatic residue present in all AAA+ unfoldases, the key residues and functions of pore-2 loops have been poorly delineated[43]. Among the ClpABC family members, the pore-2 loops are more variable in sequence and length (Fig. 4b and Extended Data Fig. 8a).

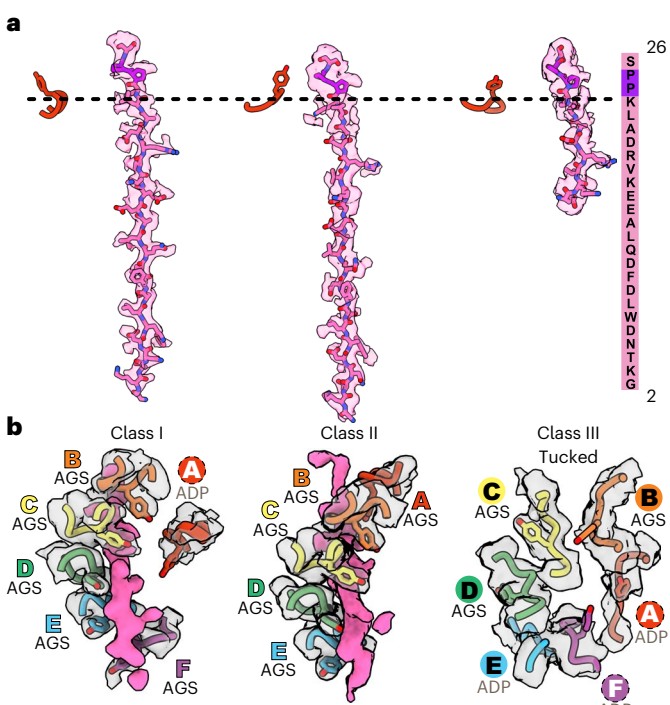

**Fig. 3 | Conformations of the ClpS NTE and D2 pore–1 loops of ClpA. a**, Density of the ClpS NTE (transparent surface at contour level 0.25 with modeled residues in stick representation) in the ClpA axial channel in representative subclasses (I, II$_b$ and III$_b$). Pro[24] and Pro[25] (colored purple) are part of the junction sequence between the NTE and the ClpS core domain. The ClpS NTE sequence is shown on the right. The D1 pore-1 loop of subunit A is shown as a reference point for the top of ClpA. **b**, Side views of D2 ring ClpA pore-1 loops (res. 538–543, transparent surfaces at contour level 0.25) and the ClpS NTE density (res. 2–15, pink surface at contour level 0.8) in classes I, II$_b$ and III$_b$. The ClpS NTE density is absent in the class-III D2 ring. D2 pore-1 loop modeled residues are shown as sticks. Subunit labels indicate the nucleotide and interaction with the ClpS NTE. Labels in colored text denote NTE engagement; the dotted circle denotes lack of NTE engagement, with Tyr[540] pointing towards the channel; labels in black text indicate the tucked conformation (Tyr[540] away from the channel). See also Extended Data Figs. 2c and 7 and Supplementary Fig. 5.

To quantify the extent of pore-1 versus pore-2 loop interactions in the D1 ring, we calculated the BSA of the ClpS−NTE interface with each class of pore loop using PISA[53]. Mirroring the pattern of D1 pore-1 loops bound to the ClpS NTE, multiple pore-2 loops made notable NTE interactions in all class I, II and III structures (Fig. 4c). The D1 pore-2 loops made substantially larger contributions to the interface with the ClpS NTE than the D2 pore-2 loops, as the BSA contributed by the D1 pore-2 loops was comparable to that from either the D1 KYR or D2 GYVG pore-1 loops (Extended Data Fig. 8b). For example, in class II$_c$, the BSA values for the D1 KYR loops range from 63 to 196 Å$^2$, the D1 pore-2 loops from 40 to 156 Å$^2$ and the D2 GYVG pore-1 loops from 101 to 179 Å$^2$. In contrast, the D2 pore-2 loops only weakly contact the NTE, as the BSA values of these interactions range from 20 to 74 Å$^2$. The difference in BSA values for D1 and D2 pore-2 loops did not appear to depend on specific side chain types of the NTE, although some D1-interacting residues, such as Lys[17] and Arg[19], had higher BSA values than others (Extended Data Fig. 9). Thus, pore loops in the D1 ring make a greater number of NTE interactions than pore loops in the D2 ring. The extensive network of NTE-engaging residues in the D1 ring suggests that it has more specific polypeptide binding/recognition 'capacity' than the D2 ring, as predicted by biochemical studies[22,39].

## D1 pore-2 loops mediate protein unfolding and remodeling

To test the functional importance of the D1 pore-2 loops, we mutated the AAS sequence (res. 295–297) to increase the bulkiness (QTQ), to mimic the pore-1 loop (KYR), to increase flexibility (GGG) or to delete this tripeptide (Δ295–297). As a defect in ClpS binding with these mutants was one reasonable hypothesis based on our structures, we first assayed the assembly of ternary ClpA$_6$•ClpS•N-degron peptide complexes using fluorescence anisotropy (Fig. 5a and Supplementary Fig. 7). Strikingly, all pore-2 loop variants maintained tight affinity for the ClpS•N-degron complexes and behaved similarly to wild-type ClpA ($^{WT}$ClpA) in the control experiment (which monitored the binary affinity of ClpA to the N-degron peptide). We then used these variants to assay ClpAPS degradation of the N-degron substrate YLFVQELA-GFP (Fig. 5b). Notably, all of the D1 pore-2-loop variants except QTQ were unable to degrade this substrate. These defects could arise from an inability to unfold or translocate YLFVQELA-GFP or from failure to transfer the YLFVQELA-GFP substrate from ClpS to ClpA.

For each variant, we then determined the ATP-hydrolysis rate of ClpA alone and in the presence of ClpP, ClpS and/or a directly recognized protein substrate. The ATPase rate serves as an indirect readout of functional ClpA assembly with its binding partners, which differentially modulate ATP hydrolysis by ClpA. For example, ClpP binding stimulates the ATPase rate of $^{WT}$ClpA approximately twofold, whereas ClpS suppresses the ATPase activity of ClpAP to a rate similar to that of ClpA alone[16,54]. All of the pore-2 variants had basal ATPase rates comparable to $^{WT}$ClpA and exhibited ATPase modulation by ClpP and ClpS that was generally similar to wild-type (Extended Data Fig. 10a). Furthermore, in the presence of the super-folder GFP substrate ($^{SF}$GFP-ssrA), which does not require ClpS for recognition and degradation, the ATPase rate of each ClpAP variant (with the exception of $^{KYR}$ClpAP) was moderately reduced during substrate processing, as expected from a previous study reporting ~20% suppression of ATP hydrolysis by GFP-ssrA[19]. We conclude, based on these studies, that our D1 pore-2 loop mutations do not grossly alter ClpA ATPase activity and are also unlikely to substantially change ClpA assembly with ClpP, ClpS or $^{SF}$GFP-ssrA.

Next, we assayed the ability of these D1 pore-2-loop variants to degrade FITC-casein, a molten-globule protein that does not require ClpS for recognition or robust ClpAP unfolding activity for degradation[32]. $^{KYR}$ClpAP degraded FITC-casein ~30% slower than $^{WT}$ClpAP, but the remaining D1 pore-2 variants degraded this substrate at roughly the wild-type rate (Extended Data Fig. 10b), indicating that recognition and translocation of this substrate are not substantially affected by the D1 pore-2 loop mutations. We then assayed the effects of the ClpA D1 pore-2 loop mutations on the steady-state kinetics of $^{SF}$GFP-ssrA degradation (Extended Data Fig. 10c). $K_M$ values for degradation of this highly stable native substrate by $^{WT}$ClpAP and the D1 pore-2 loop variants were within error, suggesting that the D1 pore-2 loops play little, if any, role in recognition of $^{SF}$GFP-ssrA. $V_{max}$ (maximum velocity) for $^{SF}$GFP-ssrA degradation was unaffected by the QTQ mutation, reduced approximately twofold by the GGG and Δ295–297 mutations and reduced approximately sixfold for the KYR mutant. Based on these results, we conclude that the D1 pore-2 loops can contribute to, but are not essential for, a reaction step after initial substrate recognition, presumably GFP unfolding, which is rate-limiting for degradation[55]. Importantly, however, these partial defects in unfolding by the GGG, Δ295–297 and KYR variants are insufficient to explain the inability of these mutants to degrade YLFVQELA-GFP when delivered by ClpS.

In comparison to FITC-casein and $^{SF}$GFP-ssrA, which are directly recognized by ClpAP, degradation of ClpS-dependent substrates require an additional protein-remodeling step; that is, ClpA must (1) remodel ClpS, to allow substrate transfer to ClpA, and then (2) unfold the N-degron substrate. Concurrently, ClpS reduces the ClpA ATPase rate, which in turn slows unfolding and translocation[5,12–16]. Therefore, ClpS should inhibit N-degron substrate degradation by the ClpA D1 pore-2 loop variants that we infer lack sufficient unfolding activity to

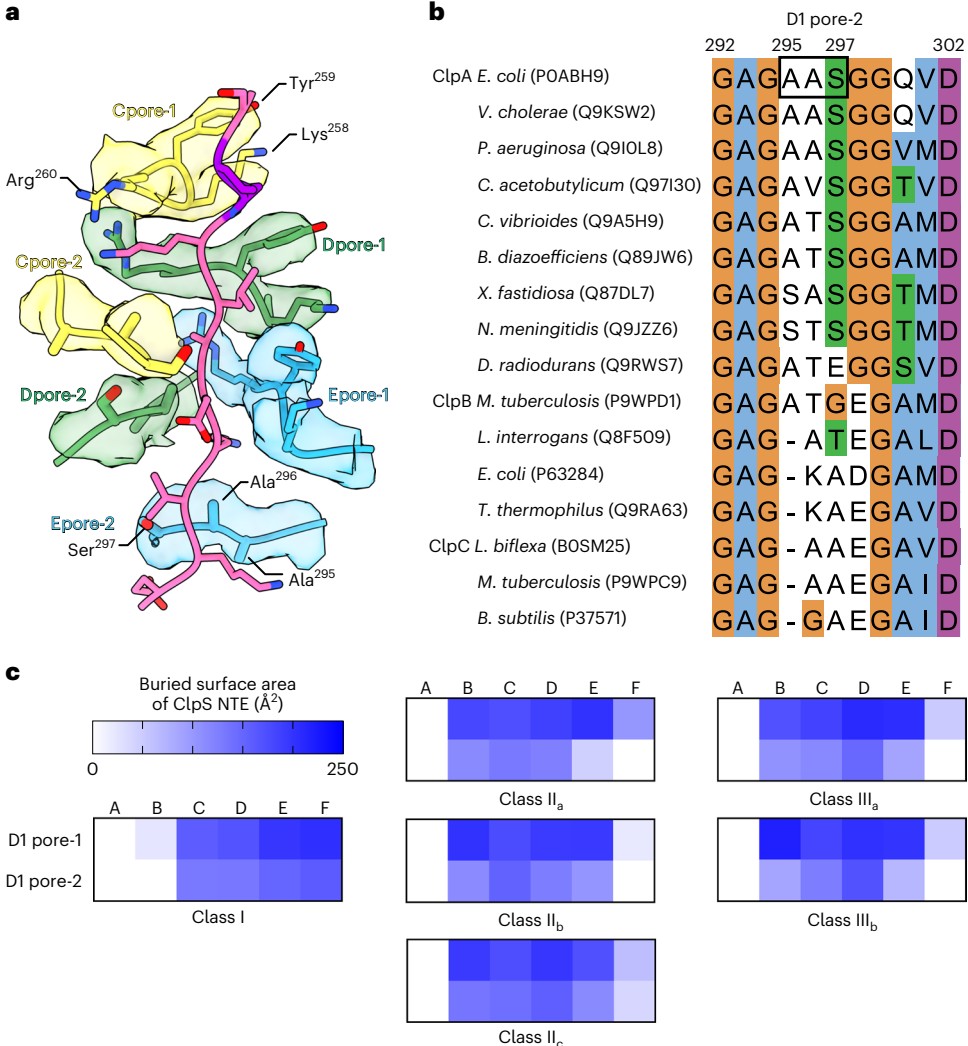

**Fig. 4 | Interaction of ClpA pore−2 loops with the ClpS NTE in the D1 ring. a**, Three pairs of pore-1 (KYR, res. 258–260) and pore-2 (AAS, res. 295–297) loops in the ClpA D1 ring of class I, shown as sticks in representative subunit coloring, and the ClpS NTE (yellow and purple sticks). The cryo-EM density of each pore loop is shown with the respective transparent surfaces at contour level 1. **b**, Multiple sequence alignment of ClpABC family members corresponding to D1 pore-2 loops of *E. coli* ClpA (res. 292–302). UniProt accession numbers are listed in parentheses. The alignment[84] at each position is colored according to ClustalX (orange, Gly; blue, hydrophobic; green, polar; magenta/purple, positive charge; white, unconserved). **c**, Buried surface area of the ClpS NTE by the pore-1 or pore-2 loops of the D1 ring in the atomic models of classes I, II and III. See also Extended Data Figs. 8 and 9.

remodel ClpS and transfer the substrate from ClpS to ClpA. We tested this hypothesis by measuring the degradation rates of YLFVQELA-GFP in the absence and presence of ClpS (Fig. 5c). Although recognition of N-end-rule substrates by ClpAP alone is intrinsically weak and normally enhanced by ClpS[56], the addition of ClpS hindered YLFVQELA-GFP degradation by the KYR, GGG and Δ295–297 variants, but not by [WT]ClpA or the QTQ variant, as predicted if the D1 pore-2 mutants are specifically defective in a ClpS remodeling step required for efficient adaptor-assisted N-degron substrate degradation.

In summary, these data suggest that the sequence identity of the AAS tripeptide (res. 295–297) alone is not critical for D1 pore-2 loop activity, as substituting these residues with QTQ had little effect on ATP hydrolysis and degradation of all substrates tested (Fig. 5b,c and Extended Data Fig. 10). Instead, changing the chemical/conformational properties of this loop by altering the charge/aromaticity (KYR) or flexibility (GGG and Δ295–297) had more profound effects. The severe defects in ClpAPS degradation conferred by the deleterious D1 pore-2 mutations but unchanged ClpS•N-degron assembly support the conclusion that D1 pore-2 loops assist in the mechanical work needed to transfer the N-degron substrate from the adaptor to the protease (and

perhaps also for subsequent reaction steps) but are not required for adaptor/substrate docking with ClpAP.

## Discussion

### The ClpS NTE is a 'degron mimic'

Our ClpAP-ClpS structures, taken with previous ClpAP structures and those of additional AAA+ family members, illustrate the variety of functional conformations AAA+ unfoldases can adopt to perform their biological functions. Importantly in all our structures, interactions between the ClpS NTE and pore loops in the ClpA channel mimic contacts observed with a polypeptide segment of the protein substrate in previous ClpA structures[23]. Specifically, the conserved tyrosines from adjacent pore-1 loops in the D1 (KYR) and D2 (GYVG) ring contact every second residue of the NTE polypeptide (Extended Data Fig. 6), with additional contacts mediated by the D1 ring pore-2 loops. Thus, in addition to its interaction with the ClpA N-domain, ClpS uses its NTE to dock tightly with the ClpA channel during substrate delivery.

Our ClpAPS structures were assembled in ATPγS, which does not fuel polypeptide translocation[36], demonstrating that binding of the entire ClpS NTE within both ClpA rings does not require this mechanical

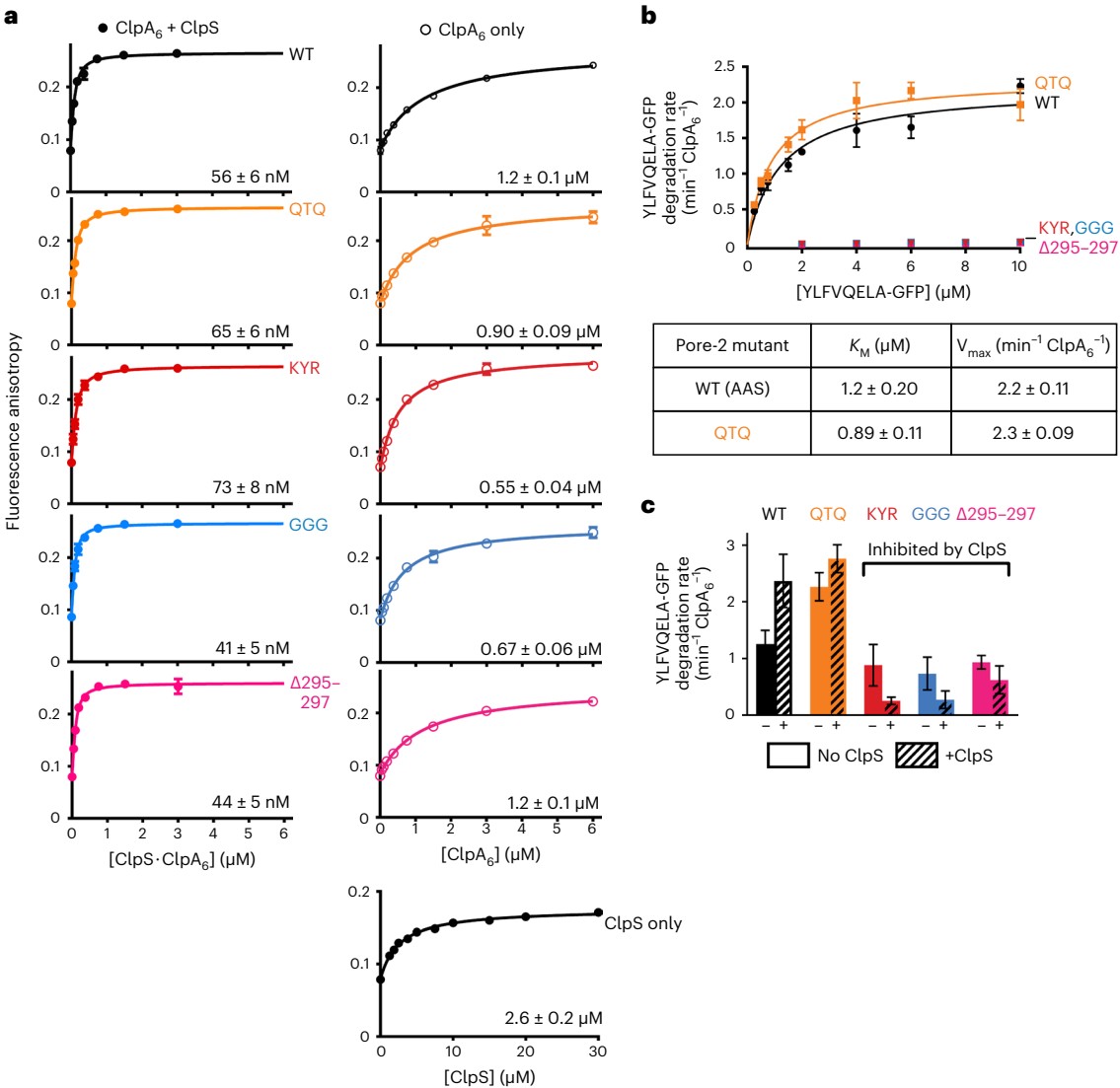

**Fig. 5 | D1 pore-2 loops are critical for ClpS-mediated degradation. a,** Fluorescence anisotropy of ClpA pore-2 variants alone (ClpA₆, open circles), with an equimolar mixture with ClpS (+ClpS, filled circles) or ClpS only (bottom right panel), titrated in increasing concentrations against a fixed concentration of fluorescein-labeled N-degron peptide. Values are mean fluorescence anisotropy values from triplicates, with error bars representing ±1 s.d., and were fit to the equations in the Methods. The $K_d$ values are reported on the lower right on each isotherm, with (±) the standard error of nonlinear least-squares and $R^2$ values = 0.99 for all fits. **b,** Kinetic analysis of YLFVQELA-GFP degradation by ClpA

D1 pore-2 variants (see Methods for concentrations). Values are mean degradation rates (min⁻¹ ClpA₆⁻¹) of triplicates, with error bars representing ±1 s.d. In the table, $K_M$ and $V_{max}$ values ± errors were obtained by nonlinear least-squares fitting to the Michaelis–Menten equation. Degradation rates of GGG, Δ295–297 and KYR could not be fit to the Michaelis–Menten equation. **c,** YLFVQELA-GFP (20 µM) degradation rates of ClpA D1 pore-2 variants (0.1 µM ClpA₆) and ClpP (0.2 µM ClpP₁₄), in the absence and presence of ClpS (0.6 µM ClpS). Summary data are mean degradation rates (min⁻¹ ClpA₆⁻¹) of triplicates, with error bars representing ±1 s.d. See Supplementary Fig. 7 for individual values for **a,b.**

activity. Together with biochemical studies and structures of substrate complexes with ATPγS-bound ClpA[23,33,57], these results suggest that any polypeptide in an unfolded/misfolded protein could passively enter an open ClpA channel, enabling ClpAP to function broadly in general protein quality control. Indeed, most of the ClpS NTE sequence is poorly conserved among orthologs and can be changed without compromising delivery of N-end-rule substrates[16], suggesting that ClpA can engage many different sequences. By contrast, the full axial channel of the ClpXP protease is blocked by a pore-2 loop before initiation of unfolding and translocation, probably limiting binding to proteins bearing highly specific ClpX degrons[50].

Our structures also reveal that ClpA pore loops bind and engage the ClpS NTE, and thus can apply mechanical force to the ClpS core domain during the N-degron delivery process. Previous studies have

demonstrated that the ClpS NTE enters the ClpA channel during assembly of delivery complexes and also can independently function as a degron for ClpAP, providing biochemical evidence that the ClpA pore loops can 'pull' on the NTE to remodel ClpS[13,14]. Blocking the ClpS NTE from entering the channel inhibits ClpS-assisted substrate degradation, reinforcing the importance of ClpA pulling on the ClpS NTE during N-degron delivery[14]. The degron-like binding of the NTE provides a structural basis for the delivery mechanism depicted in Fig. 6, in which ClpA pore loops engage the NTE, and power strokes resulting from ATP hydrolysis transmit force to mechanically remodel ClpS and thereby promote transfer of the substrate from ClpS to ClpAP for degradation.

Despite the degron-like interactions of the NTE with ClpA, ClpS is not degraded[5,13]. Interestingly, mutation of Pro²⁴–Pro²⁵ to Ala²⁴–Ala²⁵ near the ClpS NTE-core junction generates a ClpS variant that can be

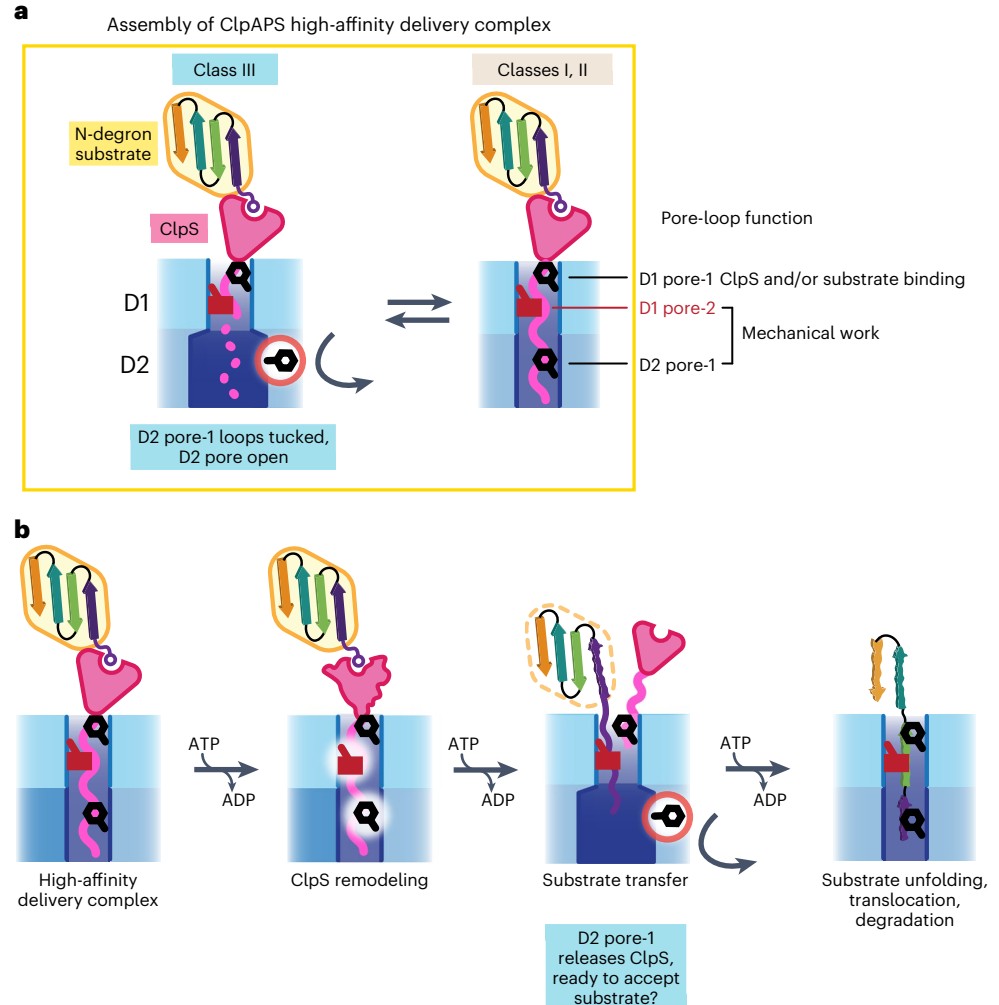

**Fig. 6 | Models of ClpS-mediated substrate binding, transfer and degradation by ClpAP. a**, Summary of observed interactions of ClpS NTE and pore-1 and pore-2 loops in D1 and D2 from cryo-EM structures. In a dynamic equilibrium (yellow box), pore-1 loops in the D2 ring (1) are tucked in and turned away from the axial channel (class III), correlated with a loss of observed ClpS NTE density, as indicated by the dotted line or (2) contact the ClpS NTE (classes I and II). Pore loop functions are indicated on the right: pore-1 loops, especially in D1, are required for ClpS and substrate binding to form delivery complexes. ATP hydrolysis powers ClpS remodeling, allowing D1 pore-2 and D2 pore-1 loops to translocate and tug on the ClpS NTE to promote substrate transfer and degradation. **b**, Proposed function of pore loops during substrate transfer. Following ClpS remodeling, D2 pore-1 loops may release the ClpS NTE, allowing for unfolding and translocation of the N-degron substrate to proceed from the D1 pore-2 loops.

degraded by ClpAP[22,58]. In our structures, Pro[24]–Pro[25] binds near the top of the ClpA channel, and the ClpS core domain is flexibly positioned directly above the pore. Although our structures only capture the initial docking of ClpS with ClpA, we propose that, during subsequent stages of ClpS delivery, ClpA pore loops may not grip Pro[24]–Pro[25] strongly enough to fully unfold ClpS, leading to 'back-slipping' in the channel and thus ClpS release. Such slipping is probably a consequence of the unique chemical properties of proline, which lacks an amide hydrogen and cannot form the extended peptide conformation adopted by the rest of the NTE in our structures and by substrate polypeptides in the channels of many AAA+ unfoldases[43]. ClpXP translocates poly-proline at an even slower rate and with a higher cost of ATP-hydrolysis cycles than poly-glycine, another homopolymer that leads to pore-loop slipping[59,60]. Other types of 'slippery' sequence adjacent to folded domains have been shown to cause release of truncated degradation products by a number of AAA+ proteases[59,61–69]. Partial ClpAP processing of native ClpS does not occur, because its NTE is not long enough to enter the ClpP peptidase chamber (Fig. 1d), but is observed for a variant bearing a duplicated NTE of ~50 residues[14].

## Implications of tucked pore-1 loops in the D2 ring

In the D2 ring of our class-III structures, many GYVG pore-1 loops assume a 'tucked' conformation in which they rotate away from the center of the ClpA channel and do not engage the ClpS NTE (Fig. 3b), presumably weakening ClpS•ClpA binding. There are several functional implications. First, the D2-disengaged/D1-engaged species could represent an intermediate in the assembly of higher-affinity ClpAPS complexes in which both rings engage the NTE (Fig. 6a). Second, tucked D2 GYVG pore loops could be important during the latter steps in ClpS-dependent substrate delivery (Fig. 6b), which require conformational remodeling of the ClpS core to weaken its interactions with the N-degron substrate and promote its transfer to ClpA[13–15]. For example, after failed attempts by ClpA to fully unfold the ClpS core, release of the NTE from the D2 ring could increase the probability that ClpS dissociates completely from ClpAP, freeing the D2 pore loops to engage the substrate for degradation (Fig. 6b). Finally, when bound to the ClpS NTE, ClpA readily adopts the class-III structures, which constitute ~20% of particles in our final dataset, suggesting pore-loop tucking is not a rare conformational state under some conditions.

More broadly, pore-loop tucking may be used during the process of enzyme pausing and/or unloading by ClpA and other AAA+ unfoldase motors. For example, the ClpA D1 ring functions as a back-up motor to prevent pausing when the principal D2-ring motor fails[20,22]. Transiently breaking contacts with the polypeptide via pore-loop tucking in only the D2 ring would allow the weaker D1 ring to continue unfolding/translocation without working against the stalled D2 motor. Subsequent untucking of the D2 pore-1 loops once the sequence causing the pause is cleared would allow the D2 motor to re-engage, restarting robust translocation by both rings. More generally, concerted loss of peptide contacts by all AAA+ domains within a ring via pore-loop rotation/tucking would be an efficient mechanism for an unfoldase either to transiently disengage from a bound polypeptide or facilitate full enzyme dissociation upon failure of a AAA+ motor to unfold, translocate or remodel a bound protein. By contrast, dissociation of an AAA+ enzyme from its polypeptide track by transitioning from a closed, substrate-bound right-handed spiral to an open, left-handed 'lock-washer' observed in some Hsp100 family members[25,26,70] requires much larger, global conformational changes throughout the AAA+ hexamer.

## Specialized functions of pore-2 loops

The pore-2 loops of other AAA+ unfoldases/remodeling enzymes have been shown to contact substrate polypeptides[23–26,43,44,48–51,71–76]. The AAS residues of the ClpA D1 pore-2 loops make substantial contacts with the ClpS NTE. Nevertheless, we find that these interactions are not critical for ClpA•ClpS binding, but instead help mediate the mechanical work needed during ClpS-assisted N-end-rule degradation (Figs. 5 and 6a). We propose that the D1 pore-2 loops of ClpA collaborate with the D2 pore-1 loops, which are also required for ClpS delivery[22], in mechanical remodeling of ClpS and/or substrate transfer to ClpA. Both sets of loops could contribute to coordinated pulling on the NTE to apply force to and remodel ClpS. Next, the pore-2 loops could capture and initiate unfolding of the N-degron substrate, and generate a sufficiently long polypeptide tail to reach the more powerful D2 pore-1 loops (Fig. 6b). Meanwhile, the D2 pore-1 loops could release the ClpS NTE via concerted loop tucking, but then untuck to grab this substrate tail for processive unfolding and translocation. Future studies parsing the interaction of pore-1 and pore-2 loops in both ClpA rings are needed to further elucidate the mechanistic steps of N-degron substrate delivery and degradation, as well as to understand why pore-2 loops are critical in ClpS-mediated degradation but less important for other classes of substrate.

## ClpA rings use both coordinated and independent action

Loss of D2 pore-1 loop engagement with the ClpS NTE is a major feature distinguishing our class-III structures from classes I and II. Although the asymmetric engagement of the NTE in the D1 but not the D2 ring of class III has some parallels with substrate-bound structures of NSF and Pex1•Pex6 (refs. [27,77,78]), the substrate-binding rings of these other enzymes adopt a 'canonical' right-hand spiral organization, whereas the ring that does not bind substrate assumes a planar conformation. In contrast, the ClpA D2 ring in class III remains in the right-handed spiral conformation. Moreover, the portion of the ClpS NTE in the D1 ring of class-III structures is bound in the same fashion as our class-I and class-II structures. This structural snapshot of divided NTE engagement between the D1 and D2 pore-1 loops reinforces biophysical and biochemical experiments that reveal a division of labor between the two AAA+ modules of ClpA[19–22]. Multiple studies of other double-ring remodeling/unfoldase enzymes, including ClpB, Hsp104, ClpC, Cdc48/ p97/VCP and the ribosomal assembly factor Rix7, report the separation of substrate binding/recognition functions in one ring from the role of the second ring as the principal motor performing mechanical work[28,41,79–83]. These results illustrate that functional specialization of individual rings is emerging as a theme shared by many double-ring AAA+ unfoldases and protein-remodeling enzymes.

## Online content

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

## Methods

### Proteins and peptides

ClpA pore-2 mutations were introduced using round-the-horn mutagenesis with T4 polynucleotide kinase and Q5 high-fidelity polymerase (New England Biolabs) into pET9a-$^{M169T}$ClpA, the plasmid used to express ClpA (gift from J. Flanagan, Hershey Medical Center). The primers used for mutagenesis are provided in Supplementary Table 1. The M169T substitution helps the overexpression of full-length ClpA[85] and is present in our laboratory version of 'wild-type' ClpA. ClpA and pore-2 variants were purified as described in ref. [16] and stored in HO buffer (50 mM HEPES-KOH, pH 7.5, 300 mM NaCl, 20 mM MgCl$_2$, 10% (wt/vol) glycerol, 0.5 mM DTT). ClpP-His$_6$ was expressed in *E. coli* strain JK10 (ref. [86]) (*clpP::cat*, Δ*lon*, *slyD::kan*, λDE3), purified as described in ref. [87] and stored in 50 mM Tris-HCl (pH 8), 150 mM KCl, 10% glycerol, 0.5 mM EDTA and 1 mM DTT. ClpS and YLFVQELA-GFP were expressed in *E. coli* strain BL21(DE3) and purified as described for His$_6$-SUMO-fusion proteins[13,16]. $^{SF}$GFP-ssrA was expressed and purified as described in ref. [88]. ClpS, $^{SF}$GFP-ssrA and YLFVQELA-GFP were stored in 25 mM HEPES-KOH, pH 7.5, 150 mM KCl, 10% glycerol and 1 mM DTT. FITC-casein (Sigma-Aldrich C0528) was dissolved in HO buffer and used freshly for biochemical assays; an extinction coefficient at 280 nm (11,460 M$^{-1}$ cm$^{-1}$) and absorbance values at 280 nm and 494 nm (to calculate and correct for overlap from the fluorescence of the FITC moiety) were used to calculate its concentration. The LLYVQRDSKEC-fluorescein N-degron synthetic peptide (21st Century Biochemicals, molecular weight 1779.9 g mol$^{-1}$) was dissolved and stored at 100 μM in 15% DMSO.

### Sample preparation and EM data acquisition

ClpA$_6$ (4 μM), ClpP$_{14}$ (8 μM), ClpS (13 μM) and YLFVQELA-GFP (13 μM) were mixed in 70 μl of assembly buffer (50 mM HEPES-KOH, pH 7.5, 300 mM KCl, 5 mM MgCl$_2$, 2 mM tris(2-carboxyethyl)phosphine, 4% glycerol, 2 mM ATPγS (Calbiochem; up to 10% ADP contamination)) for 5 min at 25 °C. A 25-μl volume of this mixture was then chromatographed at room temperature and a flow rate of 0.04 ml min$^{-1}$ on a Superdex-200 3.2/300 size-exclusion column equilibrated in assembly buffer (GE Healthcare Ettan). A 50-μl fraction containing the largest molecular weight complex was assessed by SDS–PAGE (stained with SYPRO Red; Thermo Fisher) and pooled for cryo-EM. After diluting the sample twofold in assembly buffer, a 3-μl aliquot of the mixture was applied to an R1.2/1.3 300 mesh holey-carbon gold grid (Quantifoil) glow-discharged using an easiGLOW glow-discharge device (PELCO) at 15 mA for 90 s. After a 15-s incubation, the grid was blotted for 4 s at 4 °C, 100% humidity, using Whatman grade-595 filter paper, then plunged into liquid ethane using a Vitrobot Mark IV system (Thermo Fisher Scientific).

A single grid was imaged for data collection using a Talos Arctica system with a Gatan K3 direct electron detector (University of Massachusetts Chan Medical School Cryo-EM Microscopy Facility) in super-resolution mode, operated at 200 keV. Micrographs were acquired using SerialEM 3.6 (ref. [89]) using four images with image shift per movement, with beam tilt compensation using 'Coma vs. Image Shift' and a maximum *x*-image shift of 1.8 and *y*-image shift of 4. High-resolution videos were acquired at 0.435 Å per pixel, a nominal magnification ×45,000 with a defocus range of −0.5 μm to −2.5 μm, applying a total dose of 34.71 e$^-$/Å$^2$ over 26 frames (200 ms per frame) during a 5.2-s exposure at an exposure rate of 5.1 e$^-$ per pixel per second.

### Cryo-EM data processing

Each video was binned by a factor of 2 (0.87 Å per pixel at the specimen level), aligned, corrected for beam-induced motion using MotionCor2 (ref. [90]) 1.30, then CTF estimation was calculated by CTFFIND4 (ref. [91]) 4.1.13. A total of 9,169 micrographs were analyzed using RELION-3.0.8 (ref. [92]) for data processing, classification and 3D reconstruction. The majority of auto-picked particles were doubly capped complexes

consisting of two ClpA hexamers per ClpP 14-mer. Following three rounds of 2D classification (*T* = 2, mask diameter 360 Å, no E-step limit; round 1: 200 classes, 30 iterations; round 2: 80 classes, 60 iterations; round 3: 80 classes, 60 iterations, with fast subsets), 1,043,033 particles (box size 420 pixels) were used for 3D reconstruction. The cryo-EM map of ClpAP (EMD-20406), which was collected under similar parameters as our dataset, was low-pass-filtered to 60 Å to generate an initial model for reconstruction. After the first round of 3D classification (three classes, *T* = 4, 25 iterations, no reference mask, initial low-pass filter 60 Å, mask diameter 360 Å, no E-step limit, with image alignment), two of three high-quality classes were combined, totaling 717,833 particles; the second class closely resembled the first with the exception of handedness and was flipped to correct handedness before being combined. The two classes were also utilized for per-particle CTF refinement and motion correction. The combined class had a resolution of ~3 Å. The fulcrum was shifted to the center of ClpA, and particles were re-boxed to 288 pixels to improve the resolution of this region of ClpA before performing the second round of 3D classification (six classes, *T* = 4, 100 iterations, no reference mask, initial low-pass filter 20 Å, mask diameter 180 Å, no E-step limit, with image alignment) to generate six classes. Three good classes were selected and combined (358,726 particles) for a third round of 3D classification (six classes, *T* = 20, 60 iterations, with ClpA reference mask, initial low-pass filter 8 Å, mask diameter 180 Å, no E-step limit, without image alignment) to yield the final six classes. Each class was then subjected to 3D auto-refinement without symmetry (*T* = 4, no reference mask, initial low-pass filter 8 Å, mask diameter 180 Å) to yield six maps with ~3.5 Å resolution. The directional resolution of all 3D auto-refined maps was evaluated using 3DFSC[93] 3.0. To generate the final maps, each map was density-modified and autosharpened in PHENIX[94] 1.20, giving final resolutions ranging from ~3.2 to 3.4 Å (Table 1). Low-pass-filtered cryo-EM maps were generated using EMAN2 (ref. [95]) 2.91 'e2proc3d'.

### Molecular modeling and refinement

The ClpAP cryo-EM structure (PDB 6W23) was docked into the EM map for the class-I structure, and the ClpAP cryo-EM structure (PDB 6W22) was docked into all other EM maps using 'fit in map' in Chimera[96] 1.14. Real-space refinement was performed using PHENIX with Ramachandran restraints, no secondary structure restraints and no non-crystallographic symmetry restraints, and model building was performed in Coot[97] 0.9.8. The ClpS NTE sequence (res. 2–26) was added manually in Coot. The geometry of the final models was evaluated using MolProbity[98]. EMRinger[99] and tools available in PHENIX 1.20, under 'Comprehensive Validation (cryo-EM)', were used for model-map validation. Figures and videos were generated using Chimera 1.14, ChimeraX 1.2.5 (ref. [100]) and PyMOL 2.3 (Schrödinger).

### Multiple sequence alignment

The amino acid sequences of bacterial ClpA, ClpB and ClpC proteins were downloaded from UniProtKB[101] and aligned using MUSCLE[84] with MEGA7 (ref. [102]). The sequence alignment was visualized in Jalview 1.8 (ref. [103]) and colored according to the Clustal X scheme.

### BSA calculations

The BSA of the ClpS NTE in all class structures was analyzed using the 'Protein Interfaces, Surfaces and Assemblies' service (PISA) at the European Bioinformatics Institute. (http://www.ebi.ac.uk/pdbe/prot_int/pistart.html). The BSA values were summed from the D1 pore-1 loop region (res. 254–264), D1 pore-2 loop region (res. 292–302), D2 pore-1 loop region (res. 536–544) and D2 pore-2 loop region (res. 525–531).

### Biochemical assays

Biochemical experiments were performed with at least three technical replicates at 30 °C in HO buffer using a SpectraMax M5 microplate reader (Molecular Devices) to measure the initial rates of absorbance or

fluorescence changes or equilibrium anisotropy values. ATP-hydrolysis rates were measured over the first ~2 min by monitoring the loss of absorbance at 340 nm using a coupled NADH-ATP assay[104] with 5 mM ATP (Sigma-Aldrich), pyruvate kinase (Sigma-Aldrich, P9136 at 20 U ml$^{-1}$), lactate dehydrogenase (Sigma-Aldrich, L1254 at 20 U ml$^{-1}$), 7.5 mM phosphoenolpyruvate (Sigma-Aldrich, P0564) and 0.2 mM NADH (Roche, 10107735001). ATP-hydrolysis assays were performed under four conditions: (1) ClpA$_6$ or variants (0.2 μM); (2) ClpA$_6$ or variants (0.1 μM) and ClpP$_{14}$ (0.1 μM); (3) ClpA$_6$ or variants (0.2 μM), ClpP$_{14}$ (0.2 μM) and $^{SF}$GFP-ssrA (3 μM); (4) ClpA$_6$ or variants (0.2 μM), ClpP$_{14}$ (0.2 μM) and ClpS (0.6 μM).

FITC-casein degradation assays were monitored by increases in fluorescence (excitation 340 nm, emission 520 nm) as a consequence of protease-dependent unquenching over the first 5 min; reactions contained ClpA$_6$ or variants (0.2 μM), ClpP$_{14}$ (0.4 μM), and an ATP-regeneration system (4 mM ATP, 50 μg ml$^{-1}$ creatine kinase (Sigma-Aldrich), 5 mM creatine phosphate (Sigma-Aldrich)). The endpoint fluorescence for complete FITC-casein degradation was determined by the addition of porcine elastase (100 μg ml$^{-1}$; Sigma-Aldrich) to each well, followed by a 30-min incubation before reading. To determine FITC-casein degradation rates, the increase in relative fluorescence units was normalized to the endpoint fluorescence value from fully unquenched substrate after porcine elastase incubation, and the background rate was subtracted from each reaction on the basis of a no-enzyme buffer-only control.

Degradation of GFP variants was monitored by loss of fluorescence (excitation 467 nm, emission 511 nm) over the first 5–10 min. Briefly, rates were calculated by normalizing the slope values of relative fluorescence units (RFUs)/time by the fluorescence signal determined from a standard curve of RFUs versus varying concentrations of substrate in the linear range. The degradation of different concentrations of $^{SF}$GFP-ssrA (0.25–20 μM) was assayed using ClpA$_6$ or variants (0.2 μM), ClpP$_{14}$ (0.4 μM) and the ATP-regeneration system described above. Degradation of different concentrations of YLFVQELA-GFP (0.25–20 μM) was assayed using ClpA$_6$ or variants (0.1 μM), ClpP$_{14}$ (0.2 μM), ClpS (0.6 μM) and the ATP-regeneration system. For degradation of YLFVQELA-GFP by the GGG, KYR and Δ295–297 pore-2-loop mutants shown in Fig. 5b, concentrations were ClpA$_6$ variant (0.6 μM), ClpP$_{14}$ (1.2 μM) and ClpS (3.6 μM).

The binding of the peptide LLYVQRDSKEC-fluorescein (100 nM) to (1) ClpA$_6$ or variants (0.047–6 μM), (2) ClpS (1.25–30 μM) or (3) equimolar mixtures of ClpA$_6$ or variants and ClpS (0.047–3 μM) at equilibrium was assayed by fluorescence anisotropy (excitation 490 nm, emission 525 nm) in the presence of ATPγS (2 mM). Only ClpA$_6$•ClpS•peptide ternary and ClpA$_6$•peptide binary complexes have higher anisotropy levels (in comparison to ClpS•peptide binary complexes) as a result of the much larger molecular weight of ClpA$_6$ (~500 kDa) compared to that of ClpS (~10 kDa). Data were fit by a nonlinear least-squares algorithm to equations for ClpA$_6$ only and ClpS only experiments:

$$\text{Fluorescence anisotropy} = f_{\min} + \left( \frac{f_{\max} \times K_d}{K_d + X} \right)$$

or to a quadratic equation for tight binding for ClpA$_6$•ClpS complexes:

$$\text{Fluorescence anisotropy}$$
$$= f_{\min} + (f_{\max} - f_{\min}) \frac{(L + X + K_d) - \sqrt{(L + X + K_d)^2 - 4LX}}{2L}$$

where $f_{\min}$ is the background anisotropy value, $f_{\max}$ is the maximum anisotropy value at saturated binding, $L$ is the concentration of peptide (100 nM), $K_d$ is the dissociation equilibrium constant (in nM) and $X$ is the concentration of ClpA$_6$•ClpS (in nM).

All biochemical data were analyzed in GraphPad Prism 7.

## Reporting summary

Further information on research design is available in the Nature Research Reporting Summary linked to this Article.

## Data availability

Maps have been deposited in the Electron Microscopy Data Bank (EMDB) under the following accession codes: EMD-26556 for class **I**, EMD-26554 for class **II$_a$**, EMD-26555 for class **II$_b$**, EMD-26558 for class **II$_c$**, EMD-26557 for class **III$_a$** and EMD-26559 for class **III$_b$**. Atomic models have been deposited in the Protein Data Bank (PDB) under the following accession codes: 7UIX for class **I**, 7UIV for class **II$_a$**, 7UIW for class **II$_b$**, 7UIZ for class **II$_c$**, 7UIY for class **III$_a$** and 7UJ0 for class **III$_b$**. Amino acid sequences of ClpA, ClpB and ClpC proteins are publicly available on *E. coli* ClpA (P0ABH9), *V. cholerae* ClpA (Q9KSW2), *P. aeruginosa* ClpA (Q9I0L8), *C. acetobutylicum* ClpA (Q97I30), *C. vibriodes* ClpA (Q9A5H9), *B. diazoefficiens* ClpA (Q89JW6), *X. fastidiosa* ClpA (Q87DL7), *N. meningitidis* ClpA (Q9JZZ6), *D. radiodurans* ClpA (Q9RWS7), *M. tuberculosis* ClpB (P9WPD1), *L. interrogans* ClpB (Q8F509), *E. coli* ClpB (P63284), *T. thermophilus* ClpB (Q9RA63), *L. biflexa* ClpC (B0SM25), *M. tuberculosis* ClpC (P9WPC9) and *B. subtilis* ClpC (P37571). Previously published atomic models are available from the PDB: *E. coli* ClpS (PDB 3O2B) and for *E. coli* ClpAP•RepA-GFP Engaged1 (PDB 6W22), Disengaged (PDB 6W23) and Engaged2 (PDB 6W24). The uncropped gel shown in Supplementary Fig. 1a is provided in the Supplementary Information. The values plotted in Fig. 5 and Extended Data Fig. 10 are provided as source data. Any additional information required to reanalyze the data reported in this study is available from the corresponding author (tabaker@mit.edu). Source data are provided with this Paper.

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

## Acknowledgements

We thank E. Brignole and P. Dip for support in preparing and screening cryo-EM grids at the MIT.nano Automated Cryogenic Electric Microscopy Facility on a Talos Arctica microscope, which was a gift from the Arnold and Mabel Beckman Foundation, and C. Xu, K. Song and K. Lee for data collection at the Cryo-EM Core Facility at the University of Massachusetts Chan Medical School. We thank I. Levchenko for advice in preparing ClpAPS complexes and S. Bell, T. Bell, J. Park Morehouse, T. Shih, J. Zhang and K. Zuromski for helpful advice and feedback. This work was supported by NIH grant AI-016892 (R.T.S. and T.A.B.), the Howard Hughes Medical Institute (T.A.B.) and a National Science Foundation Graduate Research Fellowship grant (1745302, S.K.). The content is solely the responsibility of the authors and does not necessarily represent the official views of the National Institutes of Health.

## Author contributions

Conceptualization was provided by S.K., X.F., R.T.S. and T.A.B., methodology by S.K., X.F., T.A.B. and R.T.S., validation by S.K., X.F. and R.T.S., formal analysis by S.K. and X.F., investigations by S.K. and X.F., resources by S.K., writing of the original draft by S.K. and X.F., review and editing by S.K., X.F., R.T.S. and T.A.B., visualization by S.K. and X.F., supervision by R.T.S. and T.A.B. and funding acquisition by S.K., R.T.S. and T.A.B.

## Competing interests

The authors declare no competing interests.

## Additional information

**Extended data** is available for this paper at https://doi.org/10.1038/s41594-022-00850-3.

**Correspondence and requests for materials** should be addressed to Tania A. Baker.

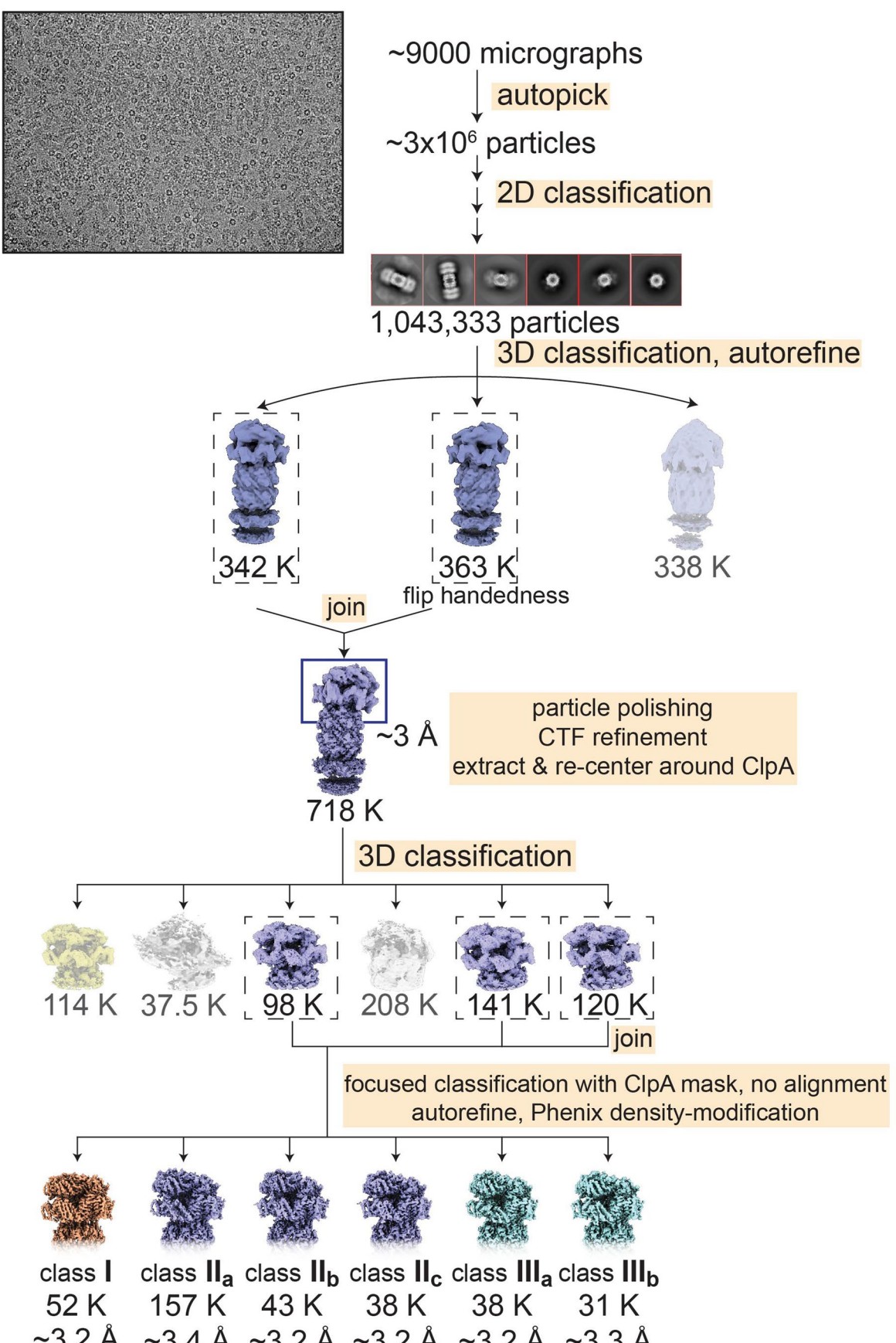

**Extended Data Fig. 1 | Cryo-EM data processing workflow diagram.** EM micrographs containing doubly-capped ClpAP complexes (two ClpA hexamers bound to one ClpP 14-mer) were processed in RELION-3. The final 3D classes were refined using density-modification in PHENIX. See also Supplementary Fig. 2.

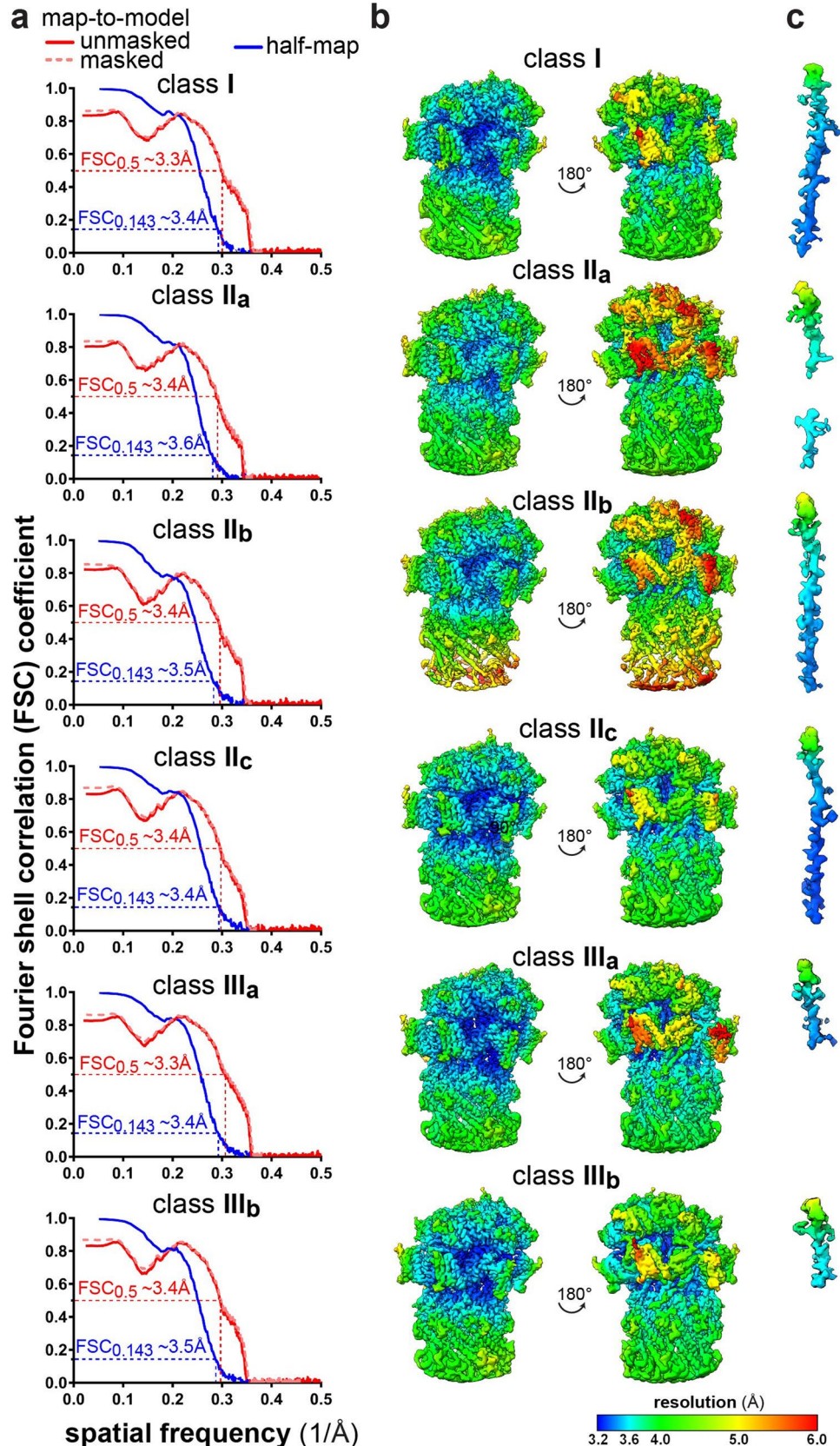

**Extended Data Fig. 2 | Cryo-EM validation. a**, Fourier Shell Correlation (FSC) plots of half maps (shown in blue) or model-map (shown in red) resolution. The dashed lines indicate the cut-off values at FSC = 0.5 (model-map) or FSC = 0.143 (half-map). **b**, Local resolution maps of final reconstructions, shown at 0.7–0.8 contour level colored according to RELION-3 calculations. **c**, Local resolution maps of ClpS NTE density shown at contour level 0.2, colored according to RELION-3 calculations.

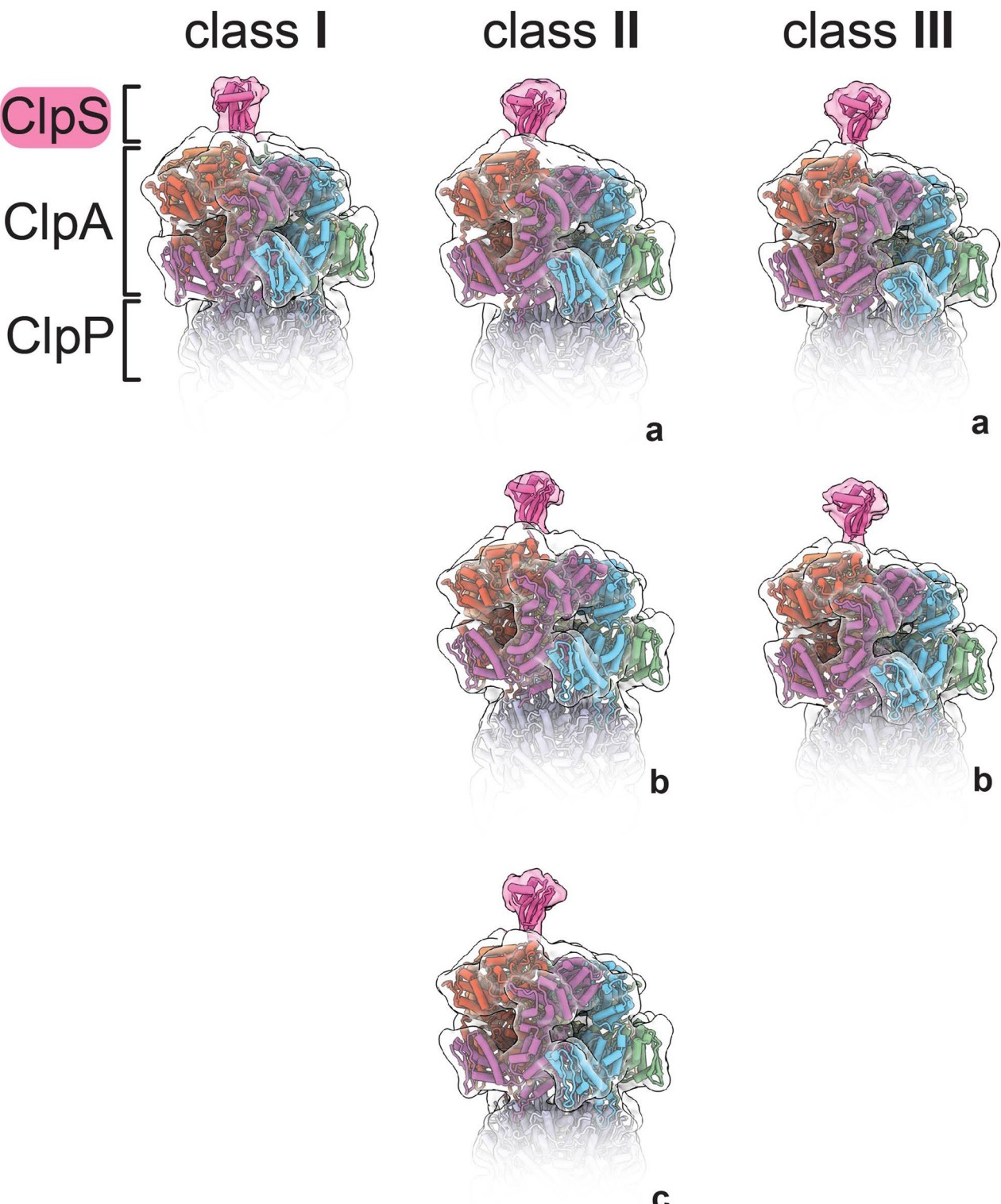

**Extended Data Fig. 3 | ClpS docking to cryo-EM maps.** The ClpS core domain (PDB 3O2B; res. 27-106) was docked to final reconstructions of class **I**, **II**$_{a-c}$, and **III**$_{a,b}$ (shown in transparent surfaces at level 0.001 of filtered maps) that were low-pass filtered to 10 Å using EMAN2 (ref. [95]) 'e2proc3d'.

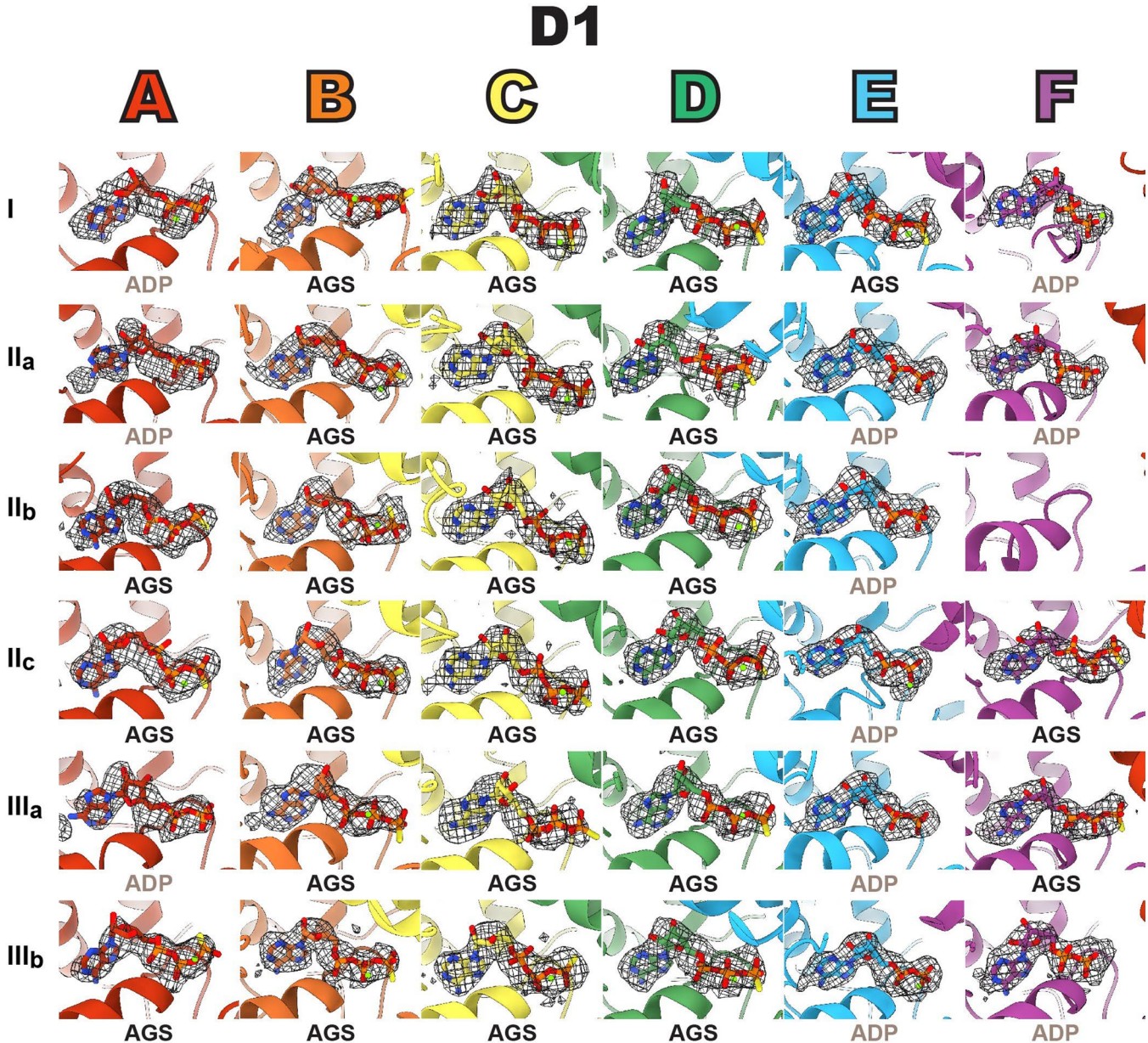

**Extended Data Fig. 4 | D1 ring nucleotide occupancy.** EM density (shown as mesh) of nucleotides in each D1 binding site of class **I**, **II_a-c**, and **III_a,b** structures. Nucleotide densities in **II_a**-A and **II_b**-A are shown at contour level 0.7. Nucleotide density in **III_a**-F is shown at contour level 1. All other nucleotide densities are shown at contour level 1.5. Nucleotide density is not observed in the D1 site of F subunit in **II_b**.

# D2

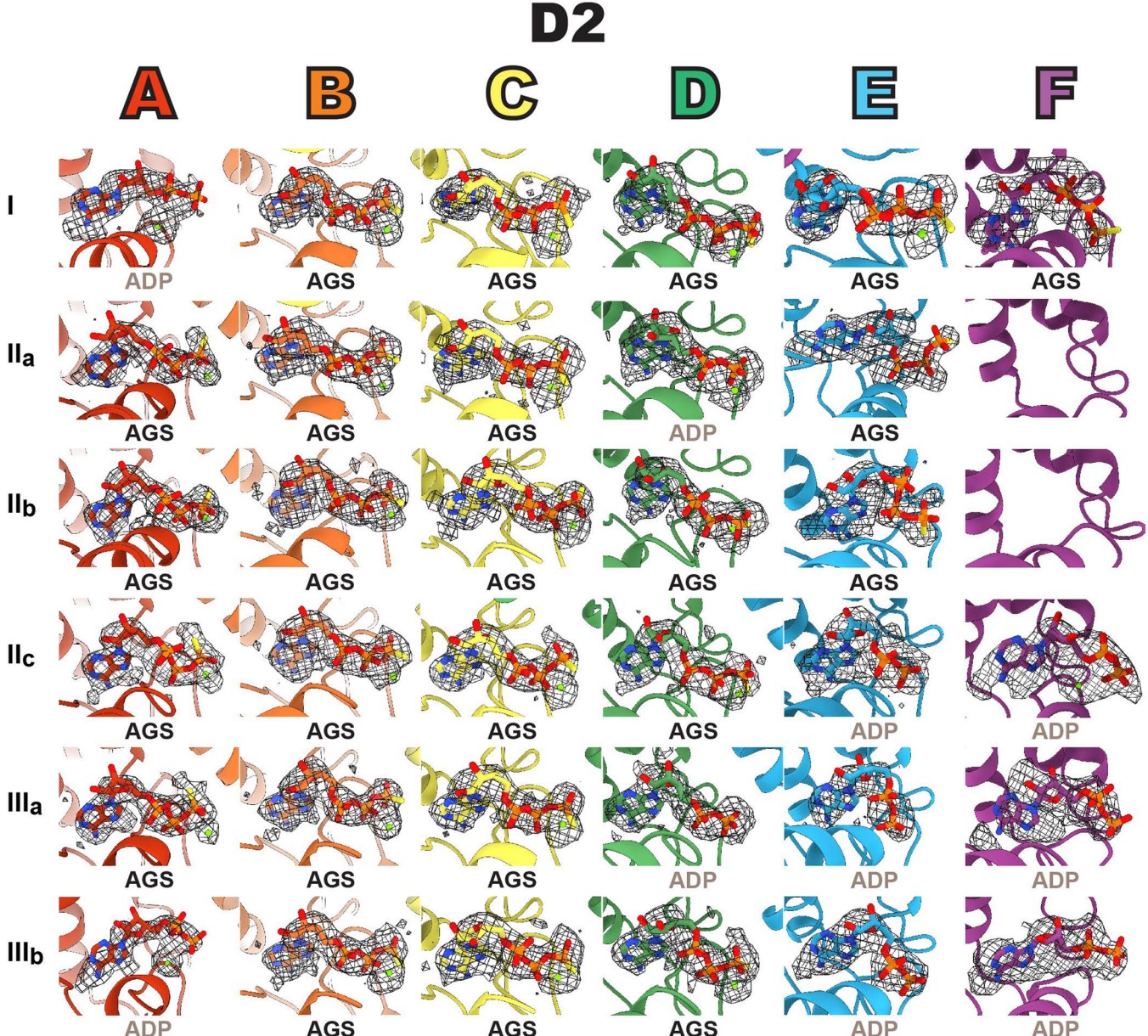

**Extended Data Fig. 5 | D2 ring nucleotide occupancy.** EM density (shown as mesh) of nucleotides in each D2 binding site of class **I**, **II**$_{a-c}$, and **III**$_{a,b}$ structures. Nucleotide densities in **I**-A, **II**$_a$-E, **II**$_c$-E, **II**$_c$-F, **III**$_a$-F, and **III**$_b$-F are shown at contour level 0.7. Nucleotide density in **I**-F is shown at contour level 1. All other nucleotide densities are shown at contour level 1.5. Nucleotide density is not observed in the D2 site of the F subunit of **II**$_a$ or **II**$_b$.

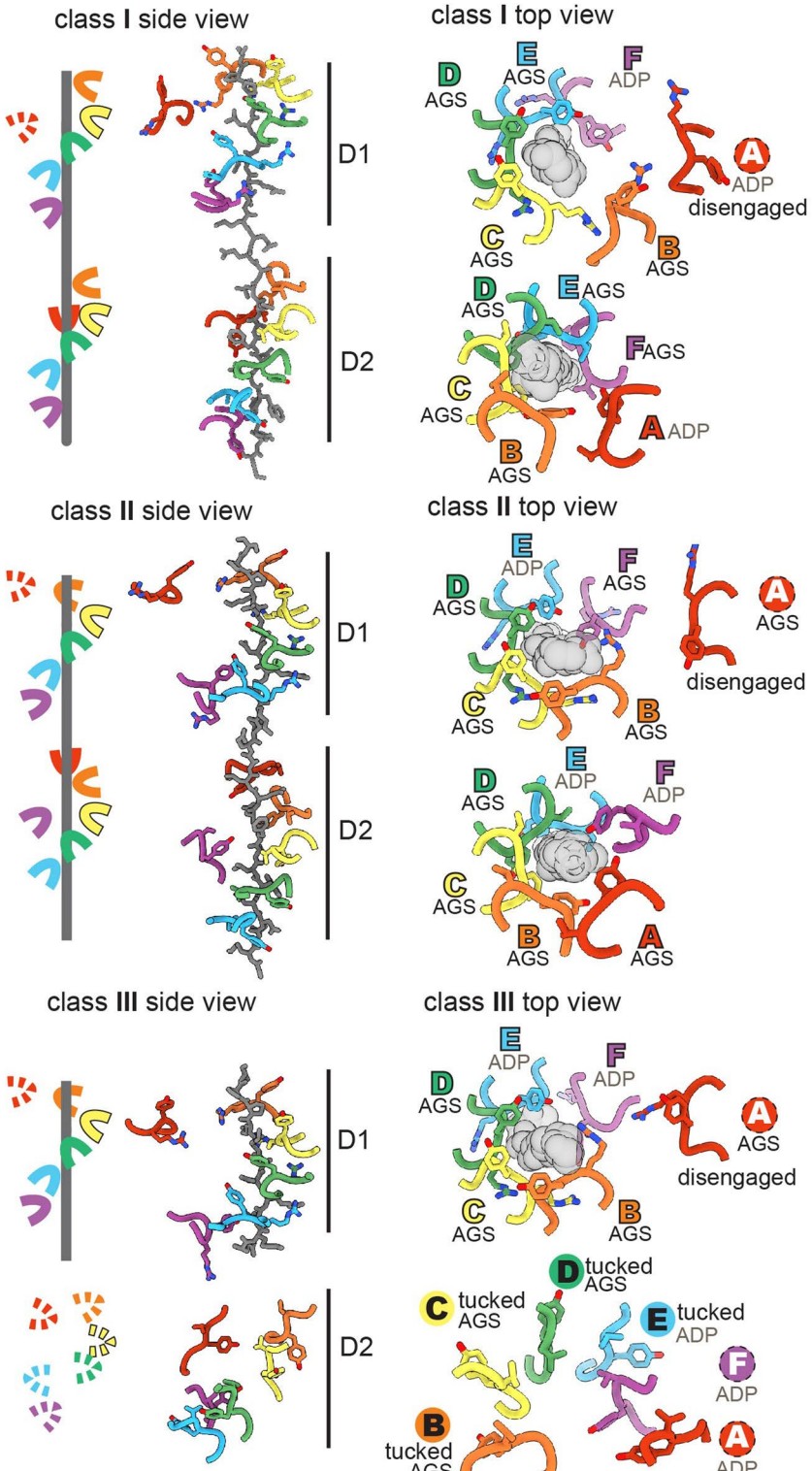

**Extended Data Fig. 6 | Pore-1 loop interactions with ClpS NTE.** The leftmost panel in each structure is a diagram of pore-1 loops in the D1 and D2 rings, where solid lines represent the presence of NTE contacts and dashed lines represent their absence. Middle panel shows lateral views of contacts between ClpA pore-1 loops and the ClpS NTE in class **I**, **II_c**, and **III_b** atomic models. The rightmost panel in each structure is a closer view of the pore-1 loops with the NTE shown as transparent spheres and the corresponding nucleotide from each subunit. Labels in colored text denote NTE engagement; the dotted circle denotes lack of NTE engagement, with Tyr$^{540}$ pointing towards the channel; labels in black text indicate the tucked conformation (Tyr$^{540}$ away from the channel).

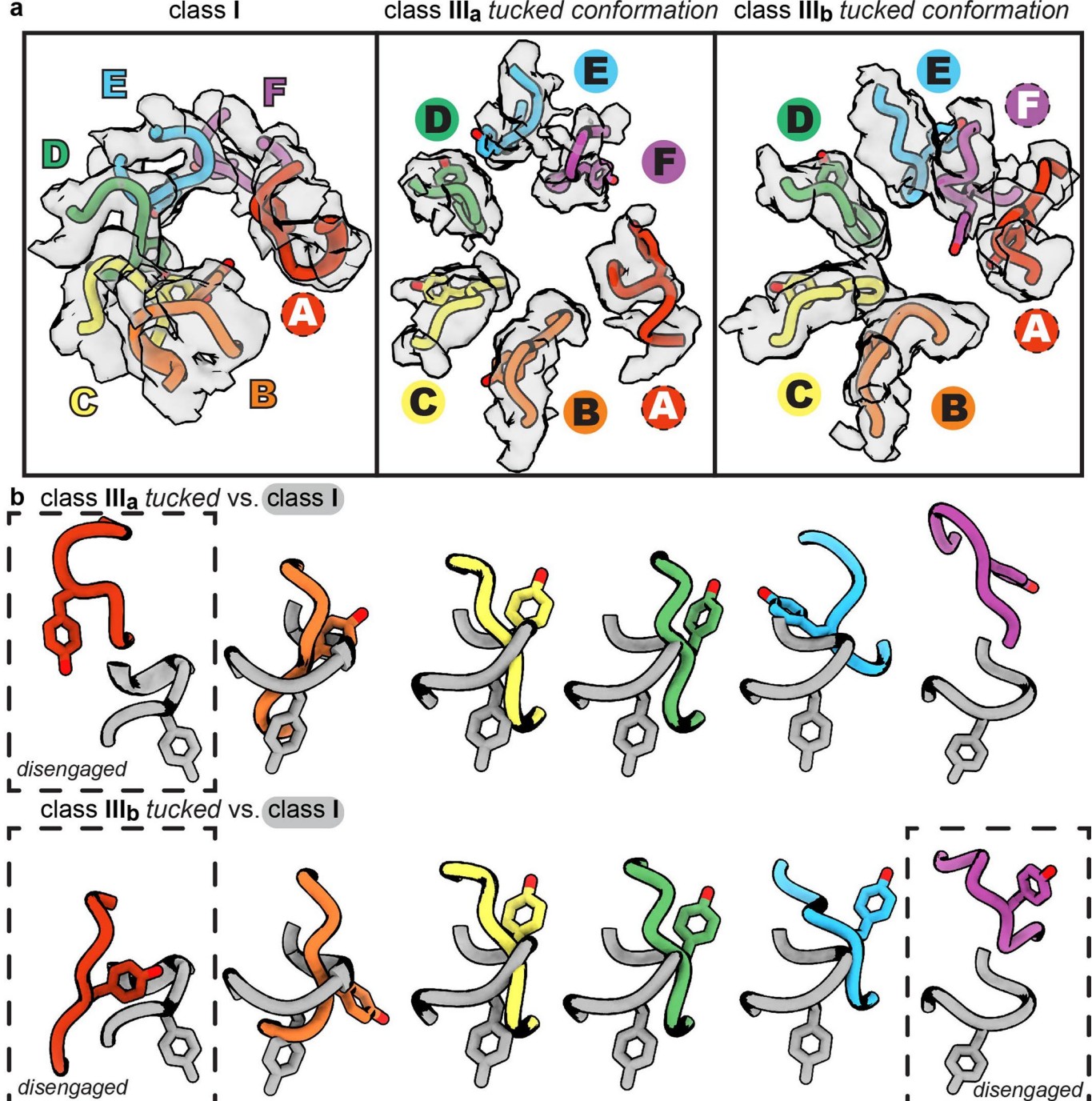

**Extended Data Fig. 7 | D2 pore-1 loop comparisons. a**, Map–model comparisons of D2 pore-1 loops (res. 538–543) in class **I** and **III** from center view of channel. Cryo-EM density is shown as transparent surfaces at contour level 0.25, and modeled residues are shown as sticks. Labels in colored text denote NTE engagement; the dotted circle denotes lack of NTE engagement, with Tyr[540] pointing towards the channel; labels in black text indicate the tucked conformation (Tyr[540] away from the channel). **b**, Atomic models of class **I** (colored gray) and **III** (respective subunit coloring) D2 pore-1 loops comparing Tyr[540] side chain orientations. Unboxed class-**III** pore loops appear to be in the tucked conformation; pore loops in the dashed box appear to be disengaged.

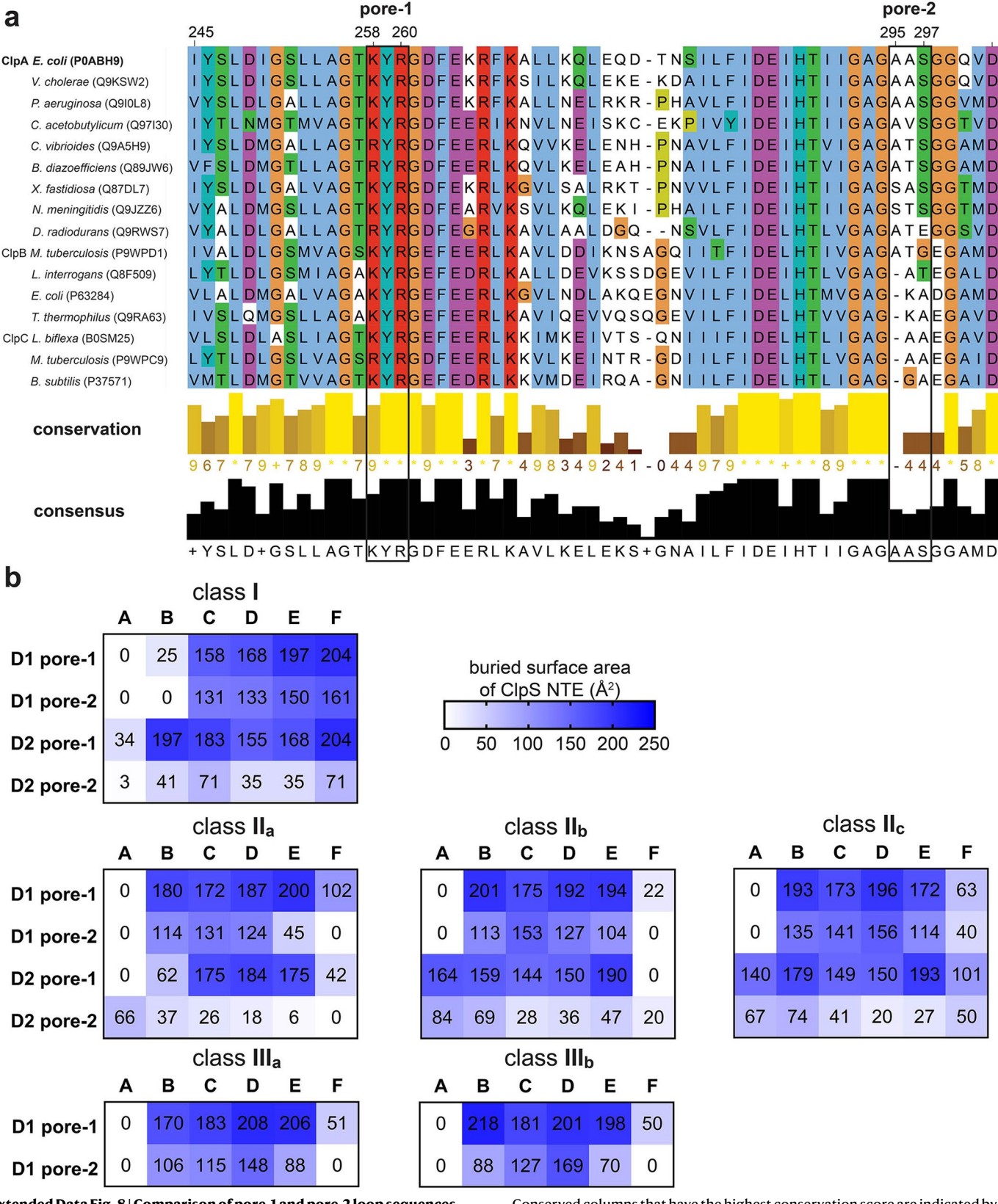

**Extended Data Fig. 8 | Comparison of pore-1 and pore-2 loop sequences and contacts to ClpS NTE. a**, Multiple sequence alignment of ClpABC family members, corresponding to pore-1 (res. 258–260, *E. coli* ClpA) and D1 pore-2 loops of *E. coli* ClpA (res. 292–302), created using MUSCLE alignment[84]. UniProt accession numbers are listed in parentheses. The alignment at each position is colored according to ClustalX (orange = Gly, blue=hydrophobic, green=polar, magenta/purple=positive charge, white=unconserved). Conservation scores are calculated in Jalview from the amino acid properties in the alignment.

Conserved columns that have the highest conservation score are indicated by '*' symbols (corresponding to a numeric score of 11), followed next by mutations that conserve all physico-chemical properties, indicated by '+' symbols. Gaps are indicated by '-', and the lowest conservation score is zero. **b**, Buried Surface Area (BSA) of ClpS NTE. Contacts between ClpS NTE and pore-1 or pore-2 loops in the D1 and D2 rings of classes **I**, **II**ₐ₋𝒸, and **III**ₐ,ᵦ were evaluated using PISA. Raw BSA values are provided in each box that correspond to coloring by the heat map scale.

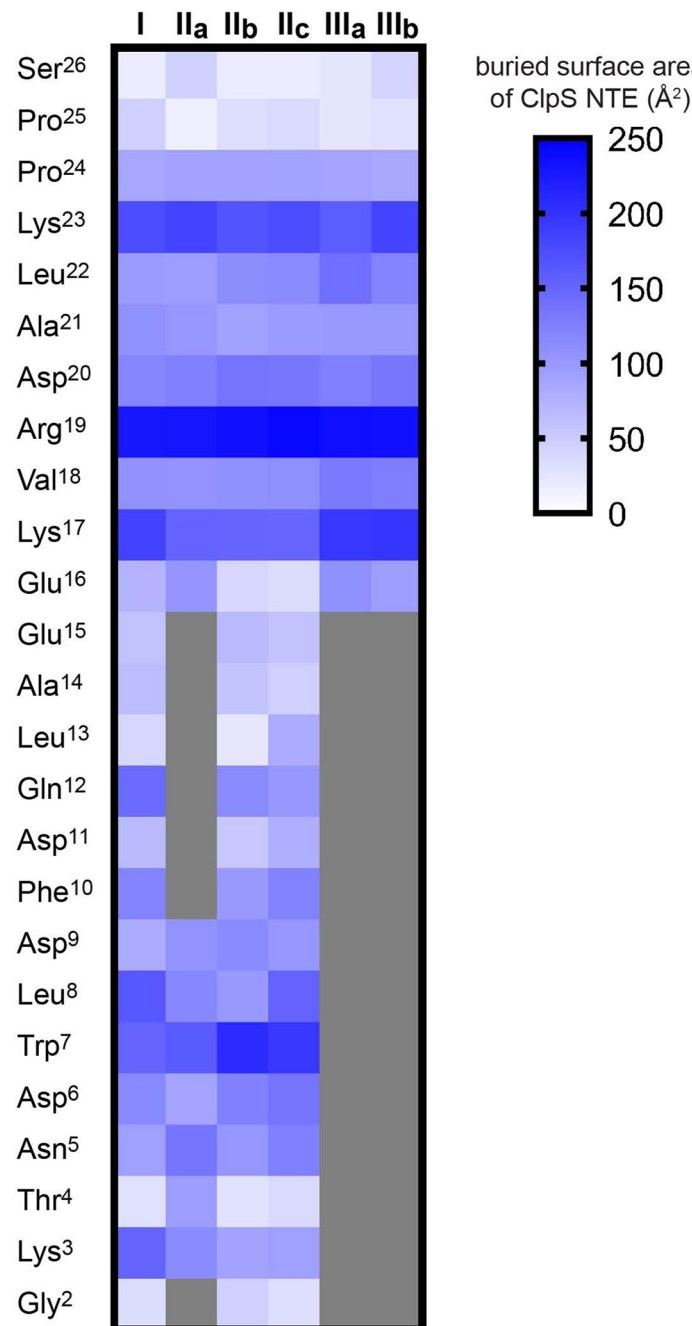

**Extended Data Fig. 9 | Total BSA of ClpS NTE.** Contacts with ClpS NTE in classes **I**, **II**$_{a-c}$, and **III**$_{a,b}$ were evaluated using PISA. BSA of NTE residues that were not modeled (gray boxes) was not calculated.

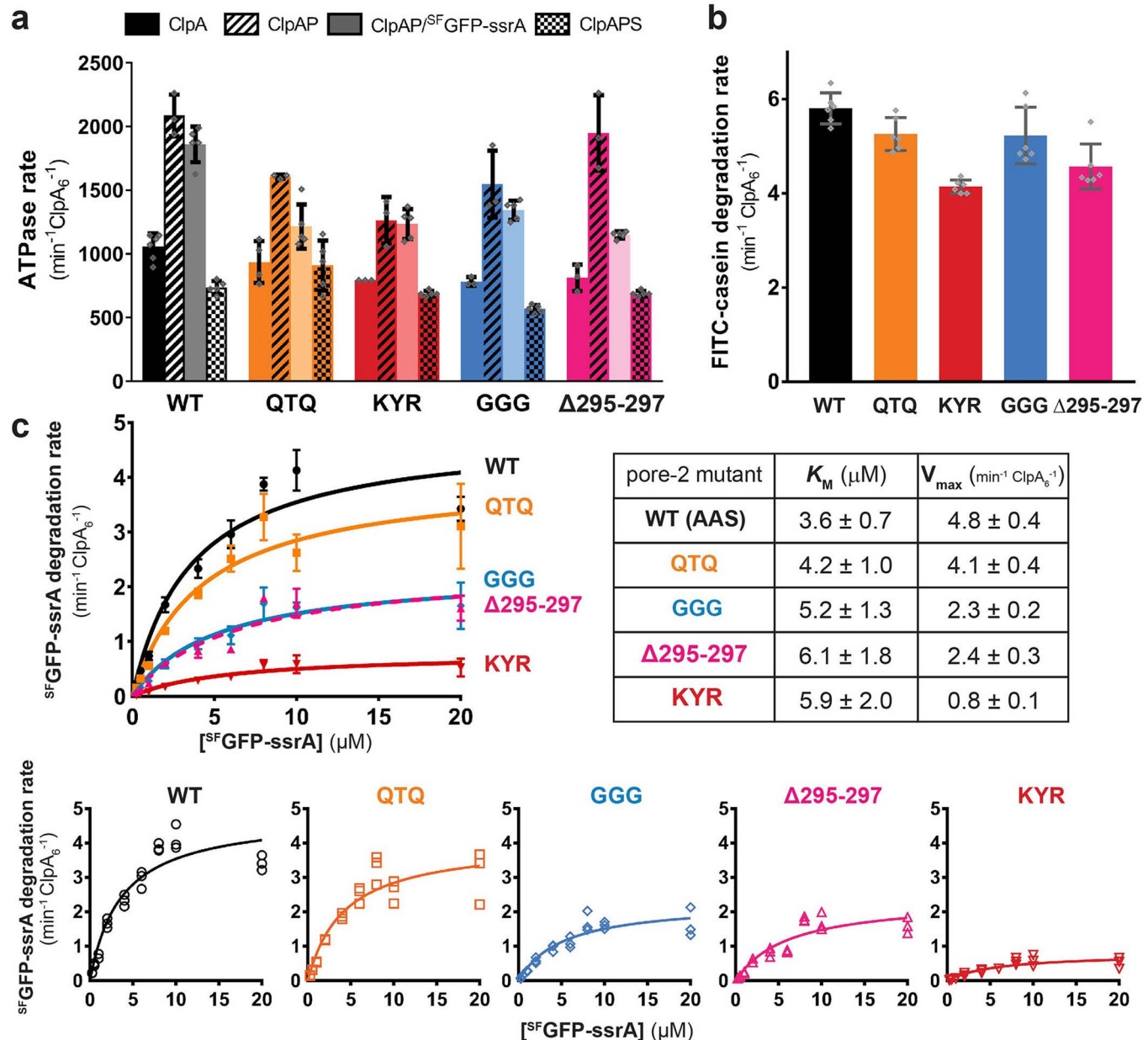

**Extended Data Fig. 10 | D1 pore-2 loop variants can support ATP hydrolysis and substrate unfolding, translocation, and degradation. a**, ATP hydrolysis rates of ClpA D1 pore-2 variants alone or in the presence of ClpP, ClpP and $^{SF}$GFP-ssrA substrate, or ClpP and ClpS; see Methods for concentrations. Bar graphs are mean ATPase rates with error bars representing ±1 S.D, $n \geq 3$ replicates. For ClpA alone, $n = 6$ (WT), $n = 4$ (QTQ), and $n = 3$ (KYR, GGG, Δ295–297). For ClpAP, $n = 3$ for all ClpA variants. For ClpAP with $^{SF}$GFP-ssrA substrate and ClpAPS, $n = 5$ for all ClpA variants. **b**, FITC-casein (20 μM) degradation rates of ClpA D1 pore-2 variants (0.2 μM ClpA$_6$) and ClpP (0.4 μM ClpP$_{14}$). Summary data are mean degradation rates (min$^{-1}$ ClpA$_6$$^{-1}$), $n = 6$ replicates with error bars representing ±1 S.D. **c**, Kinetic analysis of $^{SF}$GFP-ssrA degradation by ClpA D1 pore-2 variants (0.2 μM ClpA$_6$, 0.4 μM ClpP$_{14}$). Values are mean degradation rates (min$^{-1}$ ClpA$_6$$^{-1}$) of triplicates with error bars representing ±1 S.D. In the table, $K_M$ and $V_{max}$ values ± errors were obtained by non-linear least squares fitting to the Michaelis-Menten equation. Bottom panel shows individual degradation rates.

# Reporting Summary

## Statistics

For all statistical analyses, confirm that the following items are present in the figure legend, table legend, main text, or Methods section.

| n/a | Confirmed | |
|---|---|---|
| ☐ | ☒ | The exact sample size (*n*) for each experimental group/condition, given as a discrete number and unit of measurement |
| ☐ | ☒ | A statement on whether measurements were taken from distinct samples or whether the same sample was measured repeatedly |
| ☒ | ☐ | The statistical test(s) used AND whether they are one- or two-sided<br>*Only common tests should be described solely by name; describe more complex techniques in the Methods section.* |
| ☒ | ☐ | A description of all covariates tested |
| ☒ | ☐ | A description of any assumptions or corrections, such as tests of normality and adjustment for multiple comparisons |
| ☐ | ☒ | A full description of the statistical parameters including central tendency (e.g. means) or other basic estimates (e.g. regression coefficient) AND variation (e.g. standard deviation) or associated estimates of uncertainty (e.g. confidence intervals) |
| ☒ | ☐ | For null hypothesis testing, the test statistic (e.g. *F*, *t*, *r*) with confidence intervals, effect sizes, degrees of freedom and *P* value noted<br>*Give P values as exact values whenever suitable.* |
| ☒ | ☐ | For Bayesian analysis, information on the choice of priors and Markov chain Monte Carlo settings |
| ☒ | ☐ | For hierarchical and complex designs, identification of the appropriate level for tests and full reporting of outcomes |
| ☒ | ☐ | Estimates of effect sizes (e.g. Cohen's *d*, Pearson's *r*), indicating how they were calculated |

*Our web collection on statistics for biologists contains articles on many of the points above.*

## Software and code

Policy information about availability of computer code

| Data collection | Data collection was performed using SerialEM 3.6. |
|---|---|
| Data analysis | RELION 3.0.8, MotionCor2 1.30, and CTFFIND4 4.1.13 were used for cryo-EM data processing. Final maps were density-modified and autosharpened in PHENIX 1.20. For molecular modeling, Chimera "fit in map" was used for docking. Real-space refinement was performed using PHENIX 1.20, and model building was performed in Coot 0.9.8. MolProbity was used to evaluate geometry of the final models. 3DFSC 3.0 was used to evaluate directional resolution. Phenix cryo-EM validation and EMRinger were used to assess model geometry and map quality. Low-pass filtered cryo-EM maps were generated using EMAN2 2.91 "e2proc3d". Figures and movies of cryo-EM structures and models were generated in Chimera 1.14, ChimeraX 1.2.5, and PyMOL 2.3 (Schrodinger, LLC).<br><br>Buried surface area calculations were performed using the EBI PISA webserver. Axial channel diameter and length were measured using CAVER.<br><br>Amino-acid sequences of E. coli ClpA, ClpB, and ClpC proteins were aligned using MUSCLE with MEGA7, followed by visualization in Jalview 1.8.<br><br>Biochemical experiments were analyzed in GraphPad Prism 7. |

For manuscripts utilizing custom algorithms or software that are central to the research but not yet described in published literature, software must be made available to editors and reviewers. We strongly encourage code deposition in a community repository (e.g. GitHub). See the Nature Portfolio guidelines for submitting code & software for further information.

## Data

Policy information about <u>availability of data</u>

All manuscripts must include a <u>data availability statement</u>. This statement should provide the following information, where applicable:

- Accession codes, unique identifiers, or web links for publicly available datasets
- A description of any restrictions on data availability
- For clinical datasets or third party data, please ensure that the statement adheres to our <u>policy</u>

Maps were deposited in the Electron Microscopy Data Bank (EMDB) under accession codes: EMD-26556 for class I, EMD-26554 for class IIa, EMD-26555 for class IIb, EMD-26558 for class IIc, EMD-26557 for class IIIa, EMD-26559 for class IIIb. Atomic models were deposited in the Protein Data Bank (PDB) under accession codes: 7UIX for class I, 7UIV for class IIa, 7UIW for class IIb, 7UIZ for class IIc, 7UIY for class IIIa, and 7UJ0 for class IIIb.

Amino-acid sequences of ClpA, ClpB, and ClpC proteins were downloaded from UniProtKB (also provided in Fig. 4 and Ext. Data Fig. 8): E. coli ClpA (P0ABH9), V. cholerae ClpA (Q9KSW2), P. aeruginosa ClpA (Q9I0L8), C. acetobutylicum ClpA (Q97I30), C. vibrioides ClpA (Q9A5H9), B. diazoefficiens ClpA (Q89JW6), X. fastidiosa ClpA (Q87DL7), N. meningitidis ClpA (Q9JZZ6), D. radiodurans ClpA (Q9RWS7), M. tuberculosis ClpB (P9WPD1), L. interrogans ClpB (Q8F509), E. coli ClpB (P63284), T. thermophilus ClpB (Q9RA63), L. biflexa ClpC (B0SM25), M. tuberculosis ClpC (P9WPC9), and B. subtilis ClpC (P37571). Previously published atomic models are available from the PDB: E. coli ClpS (PDB 3O2B) and for E. coli ClpAP•RepA-GFP Eng1 (PDB 6W22), Dis (PDB 6W23), and Eng2 (PDB 6W24).

The uncropped gel shown in Supplementary Figure 1a is available in the Supplementary Information file. The values plotted in Fig. 5 and Ext. Data Fig. 10 are provided as Source Data online. Any additional information required to reanalyze the data reported in this study is available from the corresponding author (tabaker@mit.edu).

## Human research participants

Policy information about <u>studies involving human research participants and Sex and Gender in Research.</u>

| Reporting on sex and gender | N/A |
| --- | --- |
| Population characteristics | N/A |
| Recruitment | N/A |
| Ethics oversight | N/A |

Note that full information on the approval of the study protocol must also be provided in the manuscript.

# Field-specific reporting

Please select the one below that is the best fit for your research. If you are not sure, read the appropriate sections before making your selection.

☒ Life sciences          ☐ Behavioural & social sciences          ☐ Ecological, evolutionary & environmental sciences

For a reference copy of the document with all sections, see <u>nature.com/documents/nr-reporting-summary-flat.pdf</u>

# Life sciences study design

All studies must disclose on these points even when the disclosure is negative.

| Sample size | For biochemical experiments, n greater than or equal to 3 is sufficient, as each reaction contains thousands of molecules. The method of computing sample size for biochemical experiments is not applicable and therefore not performed, as performing biochemical experiments in biological triplicate is a widely-used replication standard.<br><br>For cryo-EM data-processing, all classes exceeded 30,000 particles.<br><br>For both biochemical experiments and cryo-EM structures, the sample sizes correspond to those previously published in our field and other structure-function studies. |
| --- | --- |
| Data exclusions | no data exclusions were used |
| Replication | Each biochemical assay was performed in at least triplicate. ClpA variants were purified and tested at least two times to ensure the activity between preps were consistent. Assay results of wild-type ClpA was benchmarked against assays with other batches of wild-type ClpA.<br><br>ClpAPS cryo-EM classes and atomic models were compared to those of a previous dataset (Lopez et. al 2020) for verification of the data processing methods used. |

| | |
|---|---|
| | All attempts at replication were successful. |
| Randomization | n/a; we did not randomize samples, as this does not apply to our in vitro studies. In essence, each aliquot of protein added is a random sample of thousands of molecules, as we cannot select individual molecules in the bulk biochemical assays used in this study. No experiments presented in this work required randomization to obtain biologically meaningful and significant results. |
| Blinding | n/a; we did not perform blinding, as this does not apply to our in vitro studies. In bulk biochemical studies, we cannot identify individual molecules to select them for analysis in our experiments. It is not possible to blind the data in this study. |

# Reporting for specific materials, systems and methods

We require information from authors about some types of materials, experimental systems and methods used in many studies. Here, indicate whether each material, system or method listed is relevant to your study. If you are not sure if a list item applies to your research, read the appropriate section before selecting a response.

## Materials & experimental systems

| n/a | Involved in the study |
|---|---|
| ☒ | Antibodies |
| ☒ | Eukaryotic cell lines |
| ☒ | Palaeontology and archaeology |
| ☒ | Animals and other organisms |
| ☒ | Clinical data |
| ☒ | Dual use research of concern |

## Methods

| n/a | Involved in the study |
|---|---|
| ☒ | ChIP-seq |
| ☒ | Flow cytometry |
| ☒ | MRI-based neuroimaging |

