## [Peer Review File · Nature Structural & Molecular Biology]

Peer Review Information

Journal: Nature Structural and Molecular Biology

Manuscript Title: AAA+ protease-adaptor structures reveal altered conformations and ring specialization

Corresponding author name(s): Professor Tania Baker

Editorial Notes:

Reviewer Comments & Decisions:

Decision Letter, initial version:

19th Apr 2022

Dear Tania,

Thank you again for submitting your manuscript "AAA+ protease-adaptor structures reveal altered conformations and ring specialization". We now have comments (below) from the 3 reviewers who evaluated your paper. In light of those reports, we remain interested in your study and would like to see your response to the comments of the referees, in the form of a revised manuscript.

I hope you will be pleased to see that the reviewers are all positive about the quality and interest of the work. However, they have specific suggestions for improving several aspects, including the presentation, analysis and discussion of the structural data. Please be sure to address/respond to all concerns of the referees in full in a point-by-point response and highlight all changes in the revised manuscript text file. If you have comments that are intended for editors only, please include those in a separate cover letter.

We are committed to providing a fair and constructive peer-review process. Do not hesitate to contact us if there are specific requests from the reviewers that you believe are technically impossible or

unlikely to yield a meaningful outcome.

We expect to see your revised manuscript within 6 weeks. If you cannot send it within this time, please contact us to discuss an extension; we would still consider your revision, provided that no similar work has been accepted for publication at NSMB or published elsewhere.

Reporting Summary:

When submitting the revised version of your manuscript, please pay close attention to our [Digital Image Integrity Guidelines](https://www.nature.com/nature-research/editorial-policies/image-integrity).

Data availability: this journal strongly supports public availability of data. All data used in accepted papers should be available via a public data repository, or alternatively, as Supplementary Information. If data can only be shared on request, please explain why in your Data Availability Statement, and also in the correspondence with your editor. Please note that for some data types, deposition in a public repository is mandatory - more information on our data deposition policies and available repositories can be found below:

<https://www.nature.com/nature-research/editorial-policies/reporting-standards#availability-of-data>

We require deposition of coordinates (and, in the case of crystal structures, structure factors) into the Protein Data Bank with the designation of immediate release upon publication (HPUB). Electron microscopy-derived density maps and coordinate data must be deposited in EMDB and released upon publication. To avoid delays in publication, dataset accession numbers must be supplied with the final accepted manuscript and appropriate release dates must be indicated at the galley proof stage.

[Redacted]

Kind regards,
Florian

Florian Ullrich, Ph.D.
Associate Editor
Nature Structural & Molecular Biology
ORCID 0000-0002-1153-2040

Referee expertise:

Referee #1: AAA proteins, structural biology

Referee #2: Clp proteases, structural biology, cryo-EM

Referee #3: AAA proteins, structural biology, cryo-EM

Reviewers' Comments:

Reviewer #1:
Remarks to the Author:
Comment

The manuscript entitled "AAA+ protease-adaptor structures reveal altered conformations and ring specialization" by Kim et al presents high-resolution cryo-EM structures of ClpAPS complexes. These structures were grouped into 3 classes and sub-classes within each class. These structures show how

ClpA pore loops interact with the ClpS N-terminal extension (NTE), which is known to be intrinsically disordered. In two classes, the NTE is bound by a spiral of pore-1 and pore-2 loops in a manner similar to substrate-polypeptide binding by many AAA+ unfoldases. Kinetic studies reveal that pore-2 loops of the ClpA D1 ring catalyze protein remodeling required for substrate delivery by ClpS. In a third class, D2 pore-1 loops are rotated, tucked away from the axial channel, and do not bind the NTE. These structures reveal conformational differences from prior ClpAP structures. Mutagenesis and biochemical experiments establish that the pore-2 loops of the ClpA D1 ring are essential for ClpS delivery of an N-degron substrate but contribute little to docking of ClpS with ClpA.

E. coli AAA unfoldase ClpAP has been used as one of the model AAA protein systems and extensively characterized. Crystal structures of ClpA, ClpP, and ClpS were among the very first AAA protein-related structures to be determined. More recently, structures of wt ClpAP in complex with an artificial RepA-GFP substrate were determined by cryo-EM to $\sim 3 \text{ \AA}$ resolution. Nucleotide-specific rearrangements in the AAA+ domains were identified that support a two amino acid-step translocation cycle. A lot have been learnt about the mechanisms of function of AAA unfoldases from recent EM studies of ClpA and other type II AAA ATPases. This work, like many others from the authors' laboratory, adds new information to the existing large body of knowledge about the ClpAP protease. That said, this reviewer has the following comments that require authors' attention in the revision.

Major comments

(1) In this work, the initial goal was to characterize the ClpAPS+N-degron substrate. Instead, the structure was determined for ClpAPS, lacking the substrate. Earlier and the current studies demonstrated the requirement of ClpS for the N-degron substrate degradation. It is interesting to see the discussion as illustrated in Fig. 6a that depicts the structures as high-affinity intermediates, which imply that the N-degron substrate should be present in these structures, even though it is not observable due to flexibility. While this reviewer agrees with the author that this is highly likely, it cannot be excluded that the possibility that these ClpAPS structures are not on the path for the degradation of N-degron substrates, because of the sub-stoichiometric nature of the substrate relative to ClpAPS complex (Fig 1b). This scenario should be discussed in light of the absence of the bound substrate in the lowpass maps.

(2) The paper reports the NTE of ClpS bound to the axial channels of D1 and D2. More importantly, it was able to assign sequence to the modeled peptide. This is significant because a previous work using a RepA-GFP substrate trapped in the channel failed to assign sequence. However, it is not clear how the sequence of the NTE was assigned, because the density with fit NTE peptide was not provided correctly and clearly, as in Figure 3a, where the figure is so small, and the contour level was not given. This reviewer thinks of a supplemental figure that an enlarged figure for each peptide should be given with density in mesh format and contour level clearly indicated. It would be even better if a stereoscopic pair is provided so that readers can see clearly how good the assigned sequence fit experimental density. The local resolutions can also be reported for this peptide in different structures.

Similar issues exist for modeling tucked pore-1 loop of D2 in Fig 3C, where better presentations of the map fitting should be provided and contour levels for density maps should be indicated. Also, in Extended Data Fig. 4, what are the contour levels to the maps that define bound nucleotides? Larger figures are needed so the reviewer can see more clearly and make judgement about the fitting of bound nucleotides. Please provide contour level for the Extended Data Figure 3.

(3) In the previously reported ClpAP+substrate structures, the designation of subunits in ClpA is based on their position in the spiral with p1 as the highest and p6 the lowest. In this paper, the designation of subunits (A, B, ..., F) seems not so clear. The way Fig. 2 depicts the designation of subunits is confusing and seems not unique, depending on how one tilts the structure. Nevertheless, in line 105: if ClpA has significant conformational changes, the author will need to report the different angle (or

slope) for class 1,2, 3. In Line 110: the authors will need to explain how they aligned and normalize the height for each subunit for the 3 classes. It looks like they aligned and normalize the height using IGL loops or D2; it's unclear.

(4) The author's observations between classes 1 and 2 based on the Fig 2 are very minor. Even less detailed descriptions were provided for differences among sub-classes, not to mention possible biological meaning or mechanistic insights associated with these classes. Based on what is provided, this reviewer believes that classes 1 and 2 are pretty much the same conformation with minor intrinsic mobility. Indeed, multiple 3D classifications did not improve resolution significantly and the entire paper treated classes 1 and 2 very much the same. In the previously reported ClpAP+substrate structures, different classes were based on binding of the IGL loops. It is not clear whether current grouping of different classes conform to the previous observations.

(5) Conformational changes are often difficult to depict, especially in the absence of a proper reference. Do the authors have a structure without ClpS bound (ClpAP alone). It would be great to compare the Pore-1 loops with and without ClpS NTE. So, we know that the "tucked" conformation is the conformation without its substrate or something else.

It would be better to have quantitative measurements to compare each domain to that of another class, such as height or distance from ClpP (relative height). So that way, readers can have some ideas what kind of conformational changes taking place. Alternatively, Figure 2c could use another figure to make overlap of D2 domains of class 1 or 2 to 3, so one can see which "domain" in subunit E moved and in which way, rigid body, bending, or the ring actually opens up more?

Similarly, Line 174, there is a need to show channel measurement for class3 and class 2/1 to confirm the channel is "wider". It's difficult to see the difference from Figure 2.

(6) The authors seem to think in the final section of the Discussion that the tucked pore-1 loops of D2 ring may be a general mechanism? However, this has only been observed in this structure where the bound ClpS NTE is not a degradable substrate. Furthermore, in Line 377, the sentence "the non-binding, ClpA D2 ring in class III remains in the right-handed spiral conformation." is misleading. The fact that the NTE is not seen in D2 in Class III could result from binding to different subunits, thus averaged out during reconstruction, a possibility that cannot be excluded. Moreover, this statement also contradicts earlier statement: in Line 113-115 where the authors have "By contrast, structures of ClpAP with RepA-GFP and ATPγS did not display this feature, suggesting that it arises as a consequence of ClpS binding."

A related question is that if the subunit E shifted due to ClpS binding, then class 1 and 2 subunit E should be shifted as well. Why not? This will need to be explained better.

(7) Authors have calculated the buried surface area (BSA) of the ClpS-NTE interface with each pore loop class and suggested that the D1 ring has a more specific polypeptide binding/recognition 'capacity' than the D2 ring. (Sentence 195-208). However, this reviewer feels that BSA also depends upon the type of amino acid present in the substrate. The position of a particular amino acid might also affect its neighboring residues' contact area. These contacts break and form new contacts faster and dynamically during substrate translocation. Does this calculated BSA play any role in binding in this above given scenario? Any comments/explanations related to this can be incorporated in an appropriate place.

Other Comments

(1) Table 1, data collection and processing, the magnification should be "nominal magnification", in

order to distinguish it from “calibrated magnification”. Also, PDB IDs seem non-conventional.

(2) For the uninitiated, Line 112, the sentence will need to be more specific. The authors will need to specify what is the small domain and how does it swing outward. For example, if the domain bends, then define the bent domain and the pivot point.

(3) Extended data figure 2, every FSC curve needs to label resolution. Model-map FSC looks higher resolution than half-maps, why? Normally, model-map FSC should report slightly worse resolution because calculated map from model does not contain any noise or uncertainty or reflect mobility of each atom. Can the authors explain how they generate the model-FSC curve, perhaps in the Materials and Methods? Since with/without mask, the FSC curves look the same, perhaps just use one or the other.

(4) Angle distribution plot is not necessary unless the data has preferred orientation. So, it can be removed (optional).

Reviewer #2:

Remarks to the Author:

In their manuscript the authors report on the cryo-EM structure of a substrate-loaded ClpAPS complex formed in the presence of ATP_gS. Both ClpA AAA+ motors (D1 and D2) are resolved but not the N-terminal domains (NTDs), the docking sites for ClpS, and not the bound N-end rule model substrate. Of the adaptor protein ClpS, only the N-terminal extension (NTE), corresponding to residues 2-26, is resolved, since this portion is engaged in the ClpA translocation channel. Visualization of the ClpS NTE inside the ClpA channel lends further support to previous biochemical evidence that the ClpS NTE engages into the ClpA channel for ClpS-dependent substrate handover. Based on their structure, the authors offer a hypothesis on how ClpS manages to escape degradation despite engagement into the ClpA channel.

While a cryo-EM structure of substrate-translocating ClpAP complex and structures of ClpS bound to the ClpA N-terminal domain have been solved previously, the structure presented in this study reveals engagement of the ClpS NTE into the ClpA channel, functioning as a degron-mimic.

The manuscript is well-written, the data is of high quality and previous work is appropriately referenced.

Comments

- Lines 286-293: The authors argue that ClpA cannot hydrolyze ATP_gS, which is not correct. ATP_gS is turned over, although at a much reduced rate. This is also evident from the fact that some of the nucleotide binding sites are occupied by ADP and not ATP_gS. While it is likely that the NTE can diffuse at least part of the way into the ClpA channel, the current data do not provide conclusive evidence that hydrolysis is not required in at least some of the sites to engage the entire ClpS NTE. It is also not clear why the authors speculate that unfolded proteins could passively enter the ClpA channel, since ClpA assembles only in the presence of ATP, and since ATP_gS is not a naturally occurring nucleotide. As soon as ATP is around, the ClpA AAA+ motors move and hydrolyze ATP.

- Coloring of ClpA subunits: the three blues are very difficult to distinguish, the same is true for pink and orange. A different choice of colors would much improve all of the structure figures.

- Figure 1: Panels c and d should be moved to the supplement in order to unclutter the figure.

- Figure 2: Panels a and b should be moved to the supplement.
- Figure 3b: The “tucking” of pore-1 loops in the D2 ring in one of the three classes is not visualized very well. Panel c is not necessary and it would be better to use the space for enlarging the elements in panel b.
- Figure 6: This figure is overcrowded and should be simplified significantly.

Reviewer #3:

Remarks to the Author:

Kim et al. present an exciting and well-rounded structural and biochemical investigation into the ClpAPS N-end-rule substrate delivery complex. The authors demonstrate formation of a stable ClpAPS complex bound to ATPgS and the N-end-rule substrate, YLFVQELA-GFP. High resolution cryo-electron microscopy structures of this ClpAPS complex were determined in various conformations associated with substrate engagement and translocation. These conformations are broadly classified into classes I, II, and III, and all have well-defined density for the ClpS N-terminal extension (NTE) bound by conserved pore loop elements within the AAA+ domains of ClpA. These structures support a model wherein the NTE of ClpS acts as a degron mimic, confirming and informing models proposed in previous biochemical and biophysical investigations. The authors importantly identify a unique conformational state (class III) where the D1 pore 1 and pore 2 loops are engaged to the NTE while the pore 1 loops of the D2 ring are observed to be unengaged from substrate, in a retracted position. This new conformation is a strong indicator that the two rings of ClpA function in an independent but coordinated fashion – a notable contribution to a growing body of work supporting this model of double-ring ATPase activity. The structural studies are complemented by well-designed biochemical and biophysical experiments, which notably support a mechanism wherein the pore 2 loops of the D1 ring are critical for ClpS remodeling and N-end-rule substrate transfer. Generally, the manuscript is well written, the figures are of high quality, and the claims and proposed models are supported by data while also explaining/confirming previous biochemical investigations. The work will significantly impact the AAA+ field and will be of interest to the broader scientific community, and is thus suitable for this journal. However, a few minor issues should be addressed:

The authors refer to all six loops being engaged in the D2 ring of classes I and IIc. While there does appear to be contacts with all six pore loops in these classes, cartoon representations in extended data figure 5 indicate that chain A and chain F still occupy what may be considered a seam position in these states. Additionally, the buried surface area for the class I chain A D2 pore 1 loops (Extended data figure 6) seems to be rather low and the interaction appears to be not yet fully engaged. Do the authors have any comments on this observation? The statement that there are 11 engaged pore loops may be unintentionally misleading.

The authors provide model/map comparisons for the retracted conformation of pore 1 loops in the D2 ring of class III structures in Fig 3B & 3C. In Fig 3C, the authors provide one example of the engaged vs. retracted pore loop conformer with representative density. Given the potential importance of this new state and the capacity for unbound pore loops to assume variable conformations, the authors should consider including a figure that more thoroughly demonstrates the EM density for all the pore loops in the retracted conformation. The side-by-side density/atomic model snapshots are not as interpretable as a figure where an atomic model is shown within semi-transparent EM density (such as Fig. 4a). Additionally, this could be coupled with information regarding the nucleotide state that each loop corresponds to (ATP vs ADP).

In the EM methods, please include the device and settings used for glow discharge of EM grids, the software used for data acquisition, as well as the mode of data collection (stage movement per image vs. multiple images with image shift per movement). If image shift was used, was beam tilt compensation implemented, and what was the maximum image shift? For image processing, all non-default parameters used for classification and refinement (mask diameters, tau-fudge, e-step, # of classes, initial low pass filter, # of iterations, etc.). A more detailed description of the atomic modeling methodology should also be included, as well as any relevant Phenix refinement parameters and any constraints (secondary structure, Ramachandran, etc.)

The supplement should include a "representative" raw micrograph from the dataset.

Since "dose" is a volumetric measurement and reported in \AA^3 , "electron exposure" (or fluence) and "exposure rate" (or flux) should be used in the methods and Table 1. "Dose-weighted" is acceptable when used in the context of motion correction.

For straightforward reference and readability, please also include in Table 1:

- Exposure rate (or flux, e-/pixel/s)
- Number of frames collected in each movie
- Automation software (EPU, SerialEM, Leginon, etc.)
- Total # of extracted particles, total considered for 3D (particles after removing junk), # of particles in final maps
- Estimated error of translations/rotations
- Resolution of unmasked and masked reconstructions at 0.5 and 0.143 FSC
- Local resolution range
- 3DFSC Sphericity value
- Map sharpening B factor (\AA^2) / (B factor Range)
- Atomic modeling refinement package(s)
- CCvolume/CCmask
- B factors of protein residues & ligands
- CaBLAM outliers (%)
- EMRinger score

I do not review anonymously, and thank the authors for publicly sharing their submitted manuscript on the bioRxiv preprint server. This practice enables others to benefit from findings presented in this research, as well as providing the authors with feedback from the community prior to completion of formal peer review. A postdoc in my lab, Jeff Mindrebo, helped with this review.

-Gabe Lander

Author Rebuttal to Initial comments

Manuscript: NSMB-A46068B

May 31, 2022

Dear Referees:

We thank you for your insightful comments and for the opportunity to improve our manuscript. We have addressed almost all of your major and minor suggestions in the main text, methods, and main table and figures, extended data figures, and supplementary figures, which have been highlighted in yellow in the revised manuscript. Please see below for our point-by-point responses in blue. Line numbers usually refer to the revised manuscript, and references to the previous manuscript are denoted with the adjective “formerly”.

Thank you for your valuable feedback and interest in our work,

Tania Baker

Reviewers' Comments:

Reviewer #1:

Remarks to the Author:

Comment

The manuscript entitled "AAA+ protease-adaptor structures reveal altered conformations and ring specialization" by Kim et al presents high-resolution cryo-EM structures of ClpAPS complexes. These structures were grouped into 3 classes and sub-classes within each class. These structures show how ClpA pore loops interact with the ClpS N-terminal extension (NTE), which is known to be intrinsically disordered. In two classes, the NTE is bound by a spiral of pore-1 and pore-2 loops in a manner similar to substrate-polypeptide binding by many AAA+ unfoldases. Kinetic studies reveal that pore-2 loops of the ClpA D1 ring catalyze protein remodeling required for substrate delivery by ClpS. In a third class, D2 pore-1 loops are rotated, tucked away from the axial channel, and do not bind the NTE. These structures reveal conformational differences from prior ClpAP structures. Mutagenesis and biochemical experiments establish that the pore-2 loops of the ClpA D1 ring are essential for ClpS delivery of an N-degron substrate but contribute little to docking of ClpS with ClpA.

E. coli AAA unfoldase ClpAP has been used as one of the model AAA protein systems and extensively characterized. Crystal structures of ClpA, ClpP, and ClpS were among the very first AAA protein-related structures to be determined. More recently, structures of wt ClpAP in complex with an artificial RepA-GFP substrate were determined by cryo-EM to ~ 3 Å resolution. Nucleotide-specific rearrangements in the AAA+ domains were identified that support a two amino acid-step translocation cycle. A lot have been learnt about the mechanisms of function of AAA unfoldases from recent EM studies of ClpA and other type II AAA ATPases. This work, like many others from the authors' laboratory, adds new information to the existing large body of

knowledge about the ClpAP protease. That said, this reviewer has the following comments that require authors' attention in the revision.

Major comments

(1) In this work, the initial goal was to characterize the ClpAPS+N-degron substrate. Instead, the structure was determined for ClpAPS, lacking the substrate. Earlier and the current studies demonstrated the requirement of ClpS for the N-degron substrate degradation. It is interesting to see the discussion as illustrated in Fig. 6a that depicts the structures as high-affinity intermediates, which imply that the N-degron substrate should be present in these structures, even though it is not observable due to flexibility. While this reviewer agrees with the author that this is highly likely, it cannot be excluded that the possibility that these ClpAPS structures are not on the path for the degradation of N-degron substrates, because of the sub-stoichiometric nature of the substrate relative to ClpAPS complex (Fig 1b). This scenario should be discussed in light of the absence of the bound substrate in the lowpass maps.

We thank the reviewer for this nuanced perspective on the N-degron substrate occupancy in our ClpAPS structures. Given that ClpS binds ClpA₆ ~9-fold weaker in the absence of ligand than in the presence of an N-degron substrate (Román-Hernández et al., 2011), the highest-affinity complexes contain both ClpS and substrate. Therefore, at the concentrations used, the ClpS-bound complexes used in our study should also be bound to the N-degron substrate. An additional explanation for the sub-stoichiometric ratio of N-degron substrate to ClpS could be due to concurrent binding of *apo* ClpS molecules to ClpA N-domains, as our complexes were prepared by adding ClpS in a ~3-fold molar excess of ClpA₆. We have included the possibility that the observed structures may not be on-pathway (lines 96–98), as we cannot exclude that the purified complexes may have lost the GFP substrate during sample preparation. However, as we note in the Discussion, our structural data are consistent with previous biochemical observations indicating that the ClpS NTE enters the ClpA channel during ClpS-assisted N-degron substrate degradation, supporting the likelihood that the observed structures are indeed on-pathway. Furthermore, DeDonatis et al. (2010) showed that ClpS dissociates slowly from ClpA₆ with a $t_{1/2}$ of ~3 min at 37 °C, far exceeding the amount of time needed to degrade a single substrate molecule and perhaps suggesting that ClpS could remain bound to ClpA before subsequent binding of a new substrate molecule.

References:

De Donatis, G. M., Singh, S. K., Viswanathan, S., & Maurizi, M. R. A single ClpS monomer is sufficient to direct the activity of the ClpA hexamer. *JBC* 285(12), 8771–8781 (2010).

Román-Hernández, G., Hou, J. Y., Grant, R. A., Sauer, R. T., & Baker, T. A. The ClpS adaptor mediates staged delivery of N-end rule substrates to the AAA+ ClpAP protease. *Molecular Cell*, 43(2), 217–228 (2011).

(2) The paper reports the NTE of ClpS bound to the axial channels of D1 and D2. More importantly, it was able to assign sequence to the modeled peptide. This is significant because a previous work using a RepA-GFP substrate trapped in the channel failed to assign sequence. However, it is not clear how the sequence of the NTE was assigned, because the density with fit NTE peptide was not provided correctly and clearly, as in Figure 3a, where the figure is so small, and the contour level was not given. This reviewer thinks of a supplemental figure that an enlarged figure for each peptide should be given with density in mesh format and contour level clearly indicated. It would be even better if a stereoscopic pair is provided so that readers can see clearly how good the assigned sequence fit experimental density. The local resolutions can also be reported for this peptide in different structures.

We thank the reviewer for the suggestions for Fig. 3 and have added Supplementary Fig. 5 showing the density in mesh format with the contour level indicated. We have streamlined Fig. 3 by including only a representative member of each class for clarity. The local density of the NTE polypeptide is also provided in Ext. Data Fig. 2C.

Similar issues exist for modeling tucked pore-1 loop of D2 in Fig 3C, where better presentations of the map fitting should be provided and contour levels for density maps should be indicated. Also, in Extended Data Fig. 4, what are the contour levels to the maps that define bound nucleotides? Larger figures are needed so the reviewer can see more clearly and make judgement about the fitting of bound nucleotides. Please provide contour level for the Extended Data Figure 3.

As requested, we have provided the contour levels for Fig. 3 and Extended Data Fig. 3–5. For all EM map densities shown in our figures, we now indicate the contour levels in each figure legend.

(3) In the previously reported ClpAP+substrate structures, the designation of subunits in ClpA is based on their position in the spiral with p1 as the highest and p6 the lowest. In this paper, the designation of subunits (A, B, ..., F) seems not so clear. The way Fig. 2 depicts the designation of subunits is confusing and seems not unique, depending on how one tilts the structure. Nevertheless, in line 105: if ClpA has significant conformational changes, the author will need to report the different angle (or slope) for class 1,2, 3. In Line 110: the authors will need to explain how they aligned and normalize the height for each subunit for the 3 classes. It looks like they aligned and normalize the height using IGL loops or D2; it's unclear.

We have clarified the subunit designation (lines 102–105) to include the description of the IGL loop of subunit E and F docking on either side of the empty ClpP cleft. Given the position of empty ClpP cleft between the clefts occupied by subunits E and F, the subunit position shown in Fig. 2 is unique. We now report the relative angle in Fig. 1C and added the alignment information for relative subunit height in the text (line 111–113) and figure legend. Specifically, for Fig. 1C, all EM maps were aligned to a common view using the ClpP₇ ring. In Figure 2A, the subunits within each spiral were aligned to the bottom of the IGL loop (residues 617–619) and the two flanking ClpP subunits (containing the ClpP cleft that binds the IGL loop) as indicated in the figure legend and the text.

(4) The author's observations between classes 1 and 2 based on the Fig 2 are very minor. Even less detailed descriptions were provided for differences among sub-classes, not to mention possible biological meaning or mechanistic insights associated with these classes. Based on what is provided, this reviewer believes that classes 1 and 2 are pretty much the same conformation with minor intrinsic mobility. Indeed, multiple 3D classifications did not improve resolution significantly and the entire paper treated classes 1 and 2 very much the same. In the previously reported ClpAP+substrate structures, different classes were based on binding of the IGL loops. It is not clear whether current grouping of different classes conform to the previous observations.

We appreciate Reviewer #1's comment regarding the similarities of class-I and class-II structures. To quantify the conformational differences between the class and subclass groupings, we have provided pairwise C_α RMSDs of the ClpA hexamer in Supplementary Fig. 4. Class I had the largest RMSDs across all class/subclass comparisons (ranging from ~3.1 to 4.7 Å). All subclasses in class II compared to each other had the smallest RMSDs (ranging from ~1.1 to 1.8 Å), whereas the RMSDs of class-III subclasses were generally more similar to each other than compared to those of class-II subclasses. We also performed pairwise C_α RMSD analysis with the previous ClpA cryo-EM structures bound to RepA-GFP by aligning the ClpA hexamer according to subunit spiral positions used in this manuscript (IGL loops in subunits E and F occupy ClpP clefts adjacent to ClpP cleft with subunit F at the lowest position in the spiral). For the analysis with PDB 6W24 (Engaged-2 state), in which the P1 IGL loop occupies the clockwise adjacent pocket, the empty cleft is located between subunit D and E. We found that all class-I/II/III subclasses compared to the previous structures (Lopez et al. 2020) had larger RMSD values (ranging from ~3.5 to 6.6 Å) than compared between the previous RepA-GFP-bound structures (Supplementary Fig. 4B). We did not observe an apparent movement of the IGL loop of subunit E, or the protomer P1, according to the naming convention used in Lopez et al. (2020). These RMSD analyses indicate that our class-I/II/III structures are distinct from these prior structures, and reinforces the class and subclass grouping we present in our manuscript.

(5) Conformational changes are often difficult to depict, especially in the absence of a proper reference. Do the authors have a structure without ClpS bound (ClpAP alone). It would be great

to compare the Pore-1 loops with and without ClpS NTE. So, we know that the “tucked” conformation is the conformation without its substrate or something else.

Neither we nor anyone else to our knowledge has determined a high-resolution structure of *E. coli* ClpAP with an axial channel devoid of substrate or an adaptor segment (*apo* ClpAP). Based on previous cryo-EM structures of AAA+ unfoldases/remodeling enzymes closely related to ClpA in the presence of AMP-PNP, yeast Hsp104 (PDB 5KNE) and *M. tuberculosis* ClpB (PDB 6ED3), which do not contain modeled substrate density, and the substrate-free ADP-bound *Y. pestis* Lon cryo-EM structure (PDB 6V11), we think it is likely that pore-1 loops in an *apo* ClpAP structure would be poorly resolved due to a higher degree of conformational flexibility in the axial channel when compared to structures bound to substrate or the ClpS NTE. We agree with Reviewer #1 that a high-resolution cryo-EM structure of substrate-free ClpAP with good pore-loop density would help determine whether the ‘tucked’ pore-loop conformation is unique to ClpS-bound complexes.

References for substrate-free cryo-EM structures:

Shin, M., Puchades, C., Asmita, A., Puri, N., Adjei, E., Wiseman, R. L., Karzai, A. W., & Lander, G. C. Structural basis for distinct operational modes and protease activation in AAA+ protease Lon. *Science Advances*, 6(21), eaba8404 (2020).

Yokom, A. L., Gates, S. N., Jackrel, M. E., Mack, K. L., Su, M., Shorter, J., & Southworth, D. R. Spiral architecture of the Hsp104 disaggregase reveals the basis for polypeptide translocation. *Nature Structural & Molecular Biology*, 23(9), 830–837 (2016).

Yu, H., Lupoli, T. J., Kovach, A., Meng, X., Zhao, G., Nathan, C. F., & Li, H. ATP hydrolysis-coupled peptide translocation mechanism of *Mycobacterium tuberculosis* ClpB. *Proceedings of the National Academy of Sciences of the United States of America*, 115(41), E9560–E9569 (2018).

It would be better to have quantitative measurements to compare each domain to that of another class, such as height or distance from ClpP (relative height). So that way, readers can have some ideas what kind of conformational changes taking place. Alternatively, Figure 2c could use another figure to make overlap of D2 domains of class 1 or 2 to 3, so one can see which “domain” in subunit E moved and in which way, rigid body, bending, or the ring actually opens up more?

We have updated Figure 2 to include the measurement from Asp²⁶² (reference point for top of spiral) to Leu⁶¹⁹ (bottom of IGL loop) for each subunit shown, demonstrating the differences in relative height of each subunit (Fig. 2A). We have added another panel (Fig. 2C) showing the overlay of the D2 ring AAA+ domains to better demonstrate the outward movement of the subunit-E small AAA+ domain.

Similarly, Line 174, there is a need to show channel measurement for class3 and class 2/1 to confirm the channel is “wider”. It’s difficult to see the difference from Figure 2.

We now provide a measurement of the width of the class-III and class-II axial channels, calculated using CAVER (Supplementary Fig. 6).

(6) The authors seem to think in the final section of the Discussion that the tucked pore-1 loops of D2 ring may be a general mechanism? However, this has only been observed in this structure where the bound ClpS NTE is not a degradable substrate. Furthermore, in Line 377, the sentence "the non-binding, ClpA D2 ring in class III remains in the right-handed spiral conformation." is misleading. The fact that the NTE is not seen in D2 in Class III could result from binding to different subunits, thus averaged out during reconstruction, a possibility that cannot be excluded. Moreover, this statement also contradicts earlier statement: in Line 113-115 where the authors have "By contrast, structures of ClpAP with RepA-GFP and ATPγS did not display this feature, suggesting that it arises as a consequence of ClpS binding."

We have removed the "non-binding" description of the ClpA D2 ring and agree with the reviewer that it is possible that the NTE could be bound in different subunits but not resolved due to heterogeneity. In all class-I/II/III structures from our dataset, the D2 ring adopts a right-hand spiral organization. Therefore, our comparison of the planar D2 ring in NSF and Pex1•Pex6 and the class-III right-handed spiral D2 ring does not pose a contradiction. The class-III D2 ring features both the right-handed spiraling subunit organization and broken rigid-body interface. To avoid confusion between the planar D2 ring conformation mentioned above in NSF and Pex1•Pex6 and the class I subunit organization description, we have replaced the description of the "more planar" class-I conformation with "shallower spiral" (line 112).

We apologize for the lack of clarity in lines 113–115 (formerly) regarding the broken rigid-body interface in our structures, and we have removed this overstatement in the main text.

A related question is that if the subunit E shifted due to ClpS binding, then class 1 and 2 subunit E should be shifted as well. Why not? This will need to be explained better.

We provide an explanation of the class-III-specific conformational changes in lines 174–184 and in the Discussion (lines 328–340, 373–388). The class-III subunit E movement occurs in only a subset of our structures (class III vs. classes I and II), suggesting that there may be multiple modes of ClpS-NTE binding: (i) NTE binding in both D1 and D2 as shown in classes I and II, or (ii) NTE binding only in D1 ring in class III. In lines 176–184, we note that class-III structures feature the break in the rigid-body interface (shift in subunit E), 3 ADPs in the D2 ring, and the tucked pore-1 loops. In the Discussion, we present some functional implications of class-III structures, such as representing an intermediate in the assembly of high-affinity ClpAPS delivery complexes and also being potentially featured in subsequent reaction steps of the N-degron substrate delivery mechanism (lines 331–338).

(7) Authors have calculated the buried surface area (BSA) of the ClpS–NTE interface with each

pore loop class and suggested that the D1 ring has a more specific polypeptide binding/recognition 'capacity' than the D2 ring. (Sentence 195-208). However, this reviewer feels that BSA also depends upon the type of amino acid present in the substrate. The position of a particular amino acid might also affect its neighboring residues' contact area. These contacts break and form new contacts faster and dynamically during substrate translocation. Does this calculated BSA play any role in binding in this above given scenario? Any comments/explanations related to this can be incorporated in an appropriate place.

We now provide the NTE–ClpA total BSA values by NTE residues in Extended Data Fig. 9 to address whether amino-acid side chain types are responsible for the difference in D1 vs. D2 pore-2 loops (incorporated into the main text in lines 206–209). Although some specific NTE residues, such as Lys¹⁷ and Arg¹⁹, have higher BSA scores, it did not appear that presence of these high-BSA-scoring residues correlated with positive or negative changes in contacted area with neighboring residues. For example, Val¹⁸ and Asp²⁰, which flank Arg¹⁹, have similar BSA values to the adjacent residues, Ala²¹-Leu²²-Lys²³-Pro²⁴ (Extended Data Fig. 9). Furthermore, BSA values of bulky hydrophobic amino acids (Phe¹⁰ or Trp⁷) or other positively charged amino acids (Lys³), in addition to Lys¹⁷ and Arg¹⁹, are within the same BSA range as other NTE residues. The lack of NTE BSA dependence on side-chain type is consistent with the processive translocation of AAA+ unfoldases/remodeling enzymes, which accommodate many different amino acid sequences found in substrate proteins. Our data suggest that BSA values of NTE residues are consistent with the position of each amino acid within the ClpA channel with respect to the position of the pore-1 and pore-2 loops. As expected, NTE residues near D1 or D2 pore-1 loops have high BSA scores compared to those that are not (for example: in the axial channel portion between the D1 and D2 rings).

Other Comments

(1) Table 1, data collection and processing, the magnification should be “nominal magnification”, in order to distinguish it from “calibrated magnification”. Also, PDB IDs seem non-conventional.

We thank the reviewer for this suggestion and have changed the Table 1 row label to “nominal magnification”. PDB “ID” has been replaced with “accession code”.

(2) For the uninitiated, Line 112, the sentence will need to be more specific. The authors will need to specify what is the small domain and how does it swing outward. For example, if the domain bends, then define the bent domain and the pivot point.

We now define the small and large AAA+ domains, as well as the hinge-linker (pivot point) in the main text (lines 113-115).

(3) Extended data figure 2, every FSC curve needs to label resolution. Model-map FSC looks higher resolution than half-maps, why? Normally, model-map FSC should report slightly worse

resolution because calculated map from model does not contain any noise or uncertainty or reflect mobility of each atom. Can the authors explain how they generate the model-FSC curve, perhaps in the Materials and Methods? Since with/without mask, the FSC curves look the same, perhaps just use one or the other.

FSC curves in Ext. Data Fig. 2 are now labeled with the respective resolutions at the indicated FSC cutoff values. Model-map FSC curves were generated in PHENIX listed under “Comprehensive Validation (cryo-EM)” using the full map and atomic model as input. To generate the final maps, each RELION-generated map was subjected to density modification (with `resolve_cryo_em`) and auto-sharpening in PHENIX. The higher resolution of model-map FSC curves is probably due to this maximum-likelihood density modification procedure. Documentation for “`resolve_cryo_em`” can be found here (https://phenix-online.org/documentation/reference/resolve_cryo_em.html), with additional references given below.

Jakobi, A. J., Wilmanns, M., and Sachse, C. Model-based local density sharpening of cryo-EM maps *eLife* 6:e27131 (2017).

Terwilliger, T.C., Ludtke, S.J., Read, R.J. et al. Improvement of cryo-EM maps by density modification. *Nat Methods* 17, 923–927 (2020).

(4) Angle distribution plot is not necessary unless the data has preferred orientation. So, it can be removed (optional).

As suggested, we removed the angle distribution plots in Ext. Data Fig. 2.

Reviewer #2:

Remarks to the Author:

In their manuscript the authors report on the cryo-EM structure of a substrate-loaded ClpAPS complex formed in the presence of ATP_gS. Both ClpA AAA+ motors (D1 and D2) are resolved but not the N-terminal domains (NTDs), the docking sites for ClpS, and not the bound N-end rule model substrate. Of the adaptor protein ClpS, only the N-terminal extension (NTE), corresponding to residues 2-26, is resolved, since this portion is engaged in the ClpA translocation channel. Visualization of the ClpS NTE inside the ClpA channel lends further support to previous biochemical evidence that the ClpS NTE engages into the ClpA channel for ClpS-dependent substrate handover. Based on their structure, the authors offer a hypothesis on how ClpS manages to escape degradation despite engagement into the ClpA channel. While a cryo-EM structure of substrate-translocating ClpAP complex and structures of ClpS bound to the ClpA N-terminal domain have been solved previously, the structure presented in this study reveals engagement of the ClpS NTE into the ClpA channel, functioning as a deprotonation mimic.

The manuscript is well-written, the data is of high quality and previous work is appropriately referenced.

Comments

- Lines 286-293: The authors argue that ClpA cannot hydrolyze ATP γ S, which is not correct. ATP γ S is turned over, although at a much reduced rate. This is also evident from the fact that some of the nucleotide binding sites are occupied by ADP and not ATP γ S. While it is likely that the NTE can diffuse at least part of the way into the ClpA channel, the current data do not provide conclusive evidence that hydrolysis is not required in at least some of the sites to engage the entire ClpS NTE. It is also not clear why the authors speculate that unfolded proteins could passively enter the ClpA channel, since ClpA assembles only in the presence of ATP, and since ATP γ S is not a naturally occurring nucleotide. As soon as ATP is around, the ClpA AAA+ motors move and hydrolyze ATP.

We have revised the main text and replaced with the language with “which does not fuel polypeptide translocation” (lines 290–291). We also removed the accompanying clause “does not require hydrolysis-dependent power strokes” (formerly lines 287–288) with “does not require this mechanical activity” (lines 291–292), as well as the statement (“pore-loop tucking does not require ATP hydrolysis, suggesting that under certain conditions” (formerly 337–338). Reviewer #2 is correct to point out that ClpA very slowly hydrolyzes ATP γ S, with a turnover $k_{cat} = (0.05 \pm 0.004) \text{ min}^{-1}$ (Miller and Lucius, 2014). In single turnover polypeptide translocation experiments, Miller and Lucius (2014) also showed that ClpA-catalyzed hydrolysis of ATP γ S is not sufficient for polypeptide translocation, consistent with observations from previous studies that ClpA can bind but not process substrates using this nucleotide analog (Thompson et al., 1994, Hoskins et al., 1998, Ishikawa et al., 2001, Effantin et al., 2010).

We prepared our ClpAPS•N-degron substrate complexes in the presence of 2 mM ATP γ S, incubated at room temperature for 5 min before size-exclusion chromatography in buffer containing 2 mM ATP γ S for a total of 50 minutes prior to vitrification. ClpA-catalyzed hydrolysis of ATP γ S would be negligible over this time course. Rather than ClpA-catalyzed ATP γ S hydrolysis, the source of ADP in our structures is likely to be as a contaminant in the commercially available ATP γ S we purchased and used in our study (the manufacturer’s data sheet stated up to 10% ADP contamination; this is now stated in the Methods in line 509).

We speculated that unfolded polypeptides can passively enter the ClpA channel in contrast to the situation with ClpX. In the ClpXP recognition complex with a bound ssrA degron (6WRF), the pore-2 loop of subunit A blocks the lower portion of the axial pore. Moreover, in several recently determined unpublished structures of substrate-free ClpXP (personal communication, A. Ghanbanpour), the axial channel is closed by the same pore-2 loop, even in structures with bound ATP. The closed-pore of ClpX may explain why ClpXP cannot

degrade casein. ClpAP does degrade casein, and no closed-pore ClpA structures have been observed to date.

References:

Effantin, G., Ishikawa, T., De Donatis, G. M., Maurizi, M. R. & Steven, A. C. Local and global mobility in the ClpA AAA+ chaperone detected by cryo-electron microscopy: Functional connotations. *Structure* 18, 553–562 (2010).

Hoskins, J. R., Pak, M., Maurizi, M. R. & Wickner, S. The role of the ClpA chaperone in proteolysis by ClpAP. *Proc. Natl Acad. Sci. USA* 95, 12135–40 (1998).

Ishikawa, T. et al. Translocation pathway of protein substrates in ClpAP protease. *Proc. Natl Acad. Sci. USA* 98, 4328–33 (2001).

Miller, J. M., & Lucius, A. L. ATP γ S competes with ATP for binding at Domain 1 but not Domain 2 during ClpA catalyzed polypeptide translocation. *Biophysical Chemistry*, 185, 58–69 (2014).

Thompson, M. W., Singh, S. K. & Maurizi, M. R. Processive degradation of proteins by the ATP-dependent Clp protease from *Escherichia coli*. Requirement for the multiple array of active sites in ClpP but not ATP hydrolysis. *J. Biol. Chem.* 269, 18209–18215 (1994).

- Coloring of ClpA subunits: the three blues are very difficult to distinguish, the same is true for pink and orange. A different choice of colors would much improve all of the structure figures.

We thank the reviewer for this suggestion. To improve the figures, we have changed the subunit coloring to a rainbow scheme.

- Figure 1: Panels c and d should be moved to the supplement in order to unclutter the figure.

We thank the reviewer for this suggestion for Fig. 1. To declutter the figure, we have moved panels B and C to Supplementary Fig. 1, leaving panel D (now Fig. 1B) as a reference for subunit naming.

- Figure 2: Panels a and b should be moved to the supplement.

We thank Reviewer #2 for the suggestion to streamline Fig. 2. To incorporate feedback from all reviewers, we have removed Fig. 2A (in line with Reviewer #2's suggestion to declutter the figure) and added distances to Fig. 2B (as suggested by Reviewer #1).

- Figure 3b: The “tucking” of pore-1 loops in the D2 ring in one of the three classes is not visualized very well. Panel c is not necessary and it would be better to use the space for enlarging the elements in panel b.

We thank the reviewer for this suggestion. We updated Fig. 3 accordingly and added Ext. Data Fig. 7 to better highlight the tucked pore-1 loops and to incorporate suggestions from the other reviewers.

- Figure 6: This figure is overcrowded and should be simplified significantly.

We thank the reviewer for this suggestion. We streamlined Fig. 6 and have reduced the panel numbers.

Reviewer #3:

Remarks to the Author:

Kim et al. present an exciting and well-rounded structural and biochemical investigation into the ClpAPS N-end-rule substrate delivery complex. The authors demonstrate formation of a stable ClpAPS complex bound to ATPgS and the N-end-rule substrate, YLFVQELA-GFP. High resolution cryo-electron microscopy structures of this ClpAPS complex were determined in various conformations associated with substrate engagement and translocation. These conformations are broadly classified into classes I, II, and III, and all have well-defined density for the ClpS N-terminal extension (NTE) bound by conserved pore loop elements within the AAA+ domains of ClpA. These structures support a model wherein the NTE of ClpS acts as a degron mimic, confirming and informing models proposed in previous biochemical and biophysical investigations. The authors importantly identify a unique conformational state (class III) where the D1 pore 1 and pore 2 loops are engaged to the NTE while the pore 1 loops of the D2 ring are observed to be unengaged from substrate, in a retracted position. This new conformation is a strong indicator that the two rings of ClpA function in an independent but coordinated fashion – a notable contribution to a growing body of work supporting this model of double-ring ATPase activity. The structural studies are complemented by well-designed biochemical and biophysical experiments, which notably support a mechanism wherein the pore 2 loops of the D1 ring are critical for ClpS remodeling and N-end-rule substrate transfer. Generally, the manuscript is well written, the figures are of high quality, and the claims and proposed models are supported by data while also explaining/confirming previous biochemical investigations. The work will significantly impact the AAA+ field and will be of interested to the broader scientific community, and is thus suitable for this journal. However, a few minor issues should be addressed:

The authors refer to all six loops being engaged in the D2 ring of classes I and IIc. While there does appear to be contacts with all six pore loops in these classes, cartoon representations in extended data figure 5 indicate that chain A and chain F still occupy what may be considered a seam position in these states. Additionally, the buried surface area for the class I chain A D2

pore 1 loops (Extended data figure 6) seems to be rather low and the interaction appears to be not yet fully engaged. Do the authors have any comments on this observation? The statement that there are 11 engaged pore loops may be unintentionally misleading.

We agree with Reviewer #3's assessment of the class-I chain A D2 pore-1 loop compared to those in other subunits (formerly Ext. Data Fig. 6, now called Ext. Data Fig. 8), based on the buried surface area values. We have clarified the description of these pore loops (lines 136–137) with more precise language.

The authors provide model/map comparisons for the retracted conformation of pore 1 loops in the D2 ring of class III structures in Fig 3B & 3C. In Fig 3C, the authors provide one example of the engaged vs. retracted pore loop conformer with representative density. Given the potential importance of this new state and the capacity for unbound pore loops to assume variable conformations, the authors should consider including a figure that more thoroughly demonstrates the EM density for all the pore loops in the retracted conformation. The side-by-side density/atomic model snapshots are not as interpretable as a figure where an atomic model is shown within semi-transparent EM density (such as Fig. 4a). Additionally, this could be coupled with information regarding the nucleotide state that each loop corresponds to (ATP vs ADP).

As requested, we have updated Fig. 3 to reflect the D2 ring pore-1 loop density with semi-transparent representation, as well as the information regarding the nucleotide state of each loop in this figure. We have also added Extended Data Fig. 7 to help visualize the difference between class-I/II pore loops vs. class-III tucked pore loops.

In the EM methods, please include the device and settings used for glow discharge of EM grids, the software used for data acquisition, as well as the mode of data collection (stage movement per image vs. multiple images with image shift per movement). If image shift was used, was beam tilt compensation implemented, and what was the maximum image shift? For image processing, all non-default parameters used for classification and refinement (mask diameters, tau-fudge, e-step, # of classes, initial low pass filter, # of iterations, etc.). A more detailed description of the atomic modeling methodology should also be included, as well as any relevant Phenix refinement parameters and any constraints (secondary structure, Ramachandran, etc.)

The supplement should include a "representative" raw micrograph from the dataset.

Since "dose" is a volumetric measurement and reported in Å^3 , "electron exposure" (or fluence) and "exposure rate" (or flux) should be used in the methods and Table 1. "Dose-weighted" is

acceptable when used in the context of motion correction.

For straightforward reference and readability, please also include in Table 1:

- Exposure rate (or flux, e-/pixel/s)
- Number of frames collected in each movie
- Automation software (EPU, SerialEM, Leginon, etc.)
- Total # of extracted particles, total considered for 3D (particles after removing junk), # of particles in final maps
- Estimated error of translations/rotations
- Resolution of unmasked and masked reconstructions at 0.5 and 0.143 FSC
- Local resolution range
- 3DFSC Sphericity value
- Map sharpening B factor (\AA^2) / (B factor Range)
- Atomic modeling refinement package(s)
- CCvolume/CCmask
- B factors of protein residues & ligands
- CaBLAM outliers (%)
- EMRinger score

As requested, we have added detailed information on grid preparation (lines 515–516), data acquisition/collection (lines 522–523, lines 526), image processing (lines 533–551), and atomic modeling (lines 552, lines 559–560) in the Methods section. We have also included a representative micrograph (inset in Ext. Data Fig. 1 and separately, Supplementary Fig. 2) and the requested values for Table 1 (line 409). Due to the non-uniform box dimensions of our final maps, we were unable to run 3DFSC on the final maps. Instead, to determine 3DFSC sphericity values, we used the unprocessed 3D auto-refine classes as inputs for 3DFSC (Supplementary Fig. 3). The 3DFSC sphericity values are lower for the six focused classes (fulcrum was centered to ClpA and re-boxed from 360 pixels to 288 pixels) than the parent class.

Decision Letter, first revision:

Our ref: NSMB-A46068B

11th Aug 2022

Dear Dr. Baker,

Thank you for submitting your revised manuscript "AAA+ protease-adaptor structures reveal altered conformations and ring specialization" (NSMB-A46068B). It has now been seen by the original referees and their comments are below. The reviewers find that the paper has improved in revision, and therefore we'll be happy in principle to publish it in Nature Structural & Molecular Biology, pending minor revisions to satisfy the referees' final requests and to comply with our editorial and formatting guidelines.

Sincerely,

Carolina

Carolina Perdigoto, PhD
Chief Editor
Nature Structural & Molecular Biology
orcid.org/0000-0002-5783-7106

Reviewer #2 (Remarks to the Author):

The authors have made considerable changes to the figures, which has greatly improved the manuscript. They have also provided further arguments for their hypothesis on ClpS NTE engagement. I am satisfied with the revisions and support publication of the manuscript in its current form.

Reviewer #3 (Remarks to the Author):

I am satisfied with the author's responses to my criticisms, and find no further issues,
-gabe

Decision Letter, final checks:

Our ref: NSMB-A46068B

23rd Aug 2022

Dear Dr. Baker,

Thank you for your patience as we've prepared the guidelines for final submission of your Nature Structural & Molecular Biology manuscript, "AAA+ protease-adaptor structures reveal altered conformations and ring specialization" (NSMB-A46068B). Please carefully follow the step-by-step instructions provided in the attached file, and add a response in each row of the table to indicate the changes that you have made. Please also check and comment on any additional marked-up edits we have proposed within the text. Ensuring that each point is addressed will help to ensure that your revised manuscript can be swiftly handed over to our production team.

In recognition of the time and expertise our reviewers provide to Nature Structural & Molecular Biology's editorial process, we would like to formally acknowledge their contribution to the external peer review of your manuscript entitled "AAA+ protease-adaptor structures reveal altered conformations and ring specialization". For those reviewers who give their assent, we will be publishing their names alongside the published article.

Nature Structural & Molecular Biology offers a Transparent Peer Review option for new original research manuscripts submitted after December 1st, 2019. As part of this initiative, we encourage our authors to support increased transparency into the peer review process by agreeing to have the reviewer comments, author rebuttal letters, and editorial decision letters published as a Supplementary item. When you submit your final files please clearly state in your cover letter whether or not you would like to participate in this initiative. Please note that failure to state your preference will result in delays in accepting your manuscript for publication.

Cover suggestions

As you prepare your final files we encourage you to consider whether you have any images or illustrations that may be appropriate for use on the cover of Nature Structural & Molecular Biology.

Nature Structural & Molecular Biology has now transitioned to a unified Rights Collection system which will allow our Author Services team to quickly and easily collect the rights and permissions required to publish your work. Approximately 10 days after your paper is formally accepted, you will receive an email in providing you with a link to complete the grant of rights. If your paper is eligible for Open Access, our Author Services team will also be in touch regarding any additional information that may be required to arrange payment for your article.

Please note that *Nature Structural & Molecular Biology* is a Transformative Journal (TJ). Authors may publish their research with us through the traditional subscription access route or make their paper immediately open access through payment of an article-processing charge (APC). Authors will not be required to make a final decision about access to their article until it has been accepted. [Find out more about Transformative Journals](https://www.springernature.com/gp/open-research/transformative-journals)

Authors may need to take specific actions to achieve [compliance](https://www.springernature.com/gp/open-research/funding/policy-compliance-faqs) with funder and institutional open access mandates. If your research is supported by a funder that requires immediate open access (e.g. according to [Plan S principles](https://www.springernature.com/gp/open-research/plan-s-compliance)) then you should select the gold OA route, and we will direct you to the compliant route where possible. For authors selecting the subscription publication route, the journal's standard licensing terms will need to be accepted, including [self-archiving policies](https://www.nature.com/nature-portfolio/editorial-policies/self-archiving-and-license-to-publish). Those licensing terms will supersede any other terms that the author or any third party may assert apply to any version of the manuscript.

Please use the following link for uploading these materials:
[Redacted]

Best regards,

Sophia Frank
Editorial Assistant
Nature Structural & Molecular Biology
nsmb@us.nature.com

On behalf of

Florian Ullrich, Ph.D.
Associate Editor
Nature Structural & Molecular Biology
ORCID 0000-0002-1153-2040

Reviewer #1:
None

Reviewer #2:
Remarks to the Author:
The authors have made considerable changes to the figures, which has greatly improved the manuscript. They have also provided further arguments for their hypothesis on ClpS NTE engagement. I am satisfied with the revisions and support publication of the manuscript in its current form.

Reviewer #3:
Remarks to the Author:
I am satisfied with the author's responses to my criticisms, and find no further issues,
-gabe

Final Decision Letter:

22nd Sep 2022

Dear Dr. Baker,

We are now happy to accept your revised paper "AAA+ protease-adaptor structures reveal altered conformations and ring specialization" for publication as a Article in Nature Structural & Molecular Biology.

Acceptance is conditional on the manuscript's not being published elsewhere and on there being no

announcement of this work to the newspapers, magazines, radio or television until the publication date in Nature Structural & Molecular Biology.

As soon as your article is published, you can generate your shareable link by entering the DOI of your article here: http://authors.springernature.com/share.

Corresponding authors will also receive an automated email with the shareable link

Note the policy of the journal on data deposition:

<http://www.nature.com/authors/policies/availability.html>.

Your paper will be published online soon after we receive proof corrections and will appear in print in the next available issue. You can find out your date of online publication by contacting the production team shortly after sending your proof corrections. Content is published online weekly on Mondays and Thursdays, and the embargo is set at 16:00 London time (GMT)/11:00 am US Eastern time (EST) on the day of publication. Now is the time to inform your Public Relations or Press Office about your paper, as they might be interested in promoting its publication. This will allow them time to prepare an accurate and satisfactory press release. Include your manuscript tracking number (NSMB-A46068C) and our journal name, which they will need when they contact our press office.

About one week before your paper is published online, we shall be distributing a press release to news organizations worldwide, which may very well include details of your work. We are happy for your institution or funding agency to prepare its own press release, but it must mention the embargo date and Nature Structural & Molecular Biology. If you or your Press Office have any enquiries in the meantime, please contact press@nature.com.

If you have not already done so, we strongly recommend that you upload the step-by-step protocols

used in this manuscript to the Protocol Exchange. Protocol Exchange is an open online resource that allows researchers to share their detailed experimental know-how. All uploaded protocols are made freely available, assigned DOIs for ease of citation and fully searchable through nature.com. Protocols can be linked to any publications in which they are used and will be linked to from your article. You can also establish a dedicated page to collect all your lab Protocols. By uploading your Protocols to Protocol Exchange, you are enabling researchers to more readily reproduce or adapt the methodology you use, as well as increasing the visibility of your protocols and papers. Upload your Protocols at www.nature.com/protocolexchange/. Further information can be found at www.nature.com/protocolexchange/about.

Please note that *Nature Structural & Molecular Biology* is a Transformative Journal (TJ). Authors may publish their research with us through the traditional subscription access route or make their paper immediately open access through payment of an article-processing charge (APC). Authors will not be required to make a final decision about access to their article until it has been accepted. [Find out more about Transformative Journals](https://www.springernature.com/gp/open-research/transformative-journals)

Sincerely,

Editorial Office
Nature Structural & Molecular Biology
nsm@us.nature.com

on behalf of

Dr Florian Ullrich

Associate Editor, Nature
Consulting Editor, Nature Structural & Molecular Biology
ORCID 0000-0002-1153-2040
